# Mitigating Hallucinations in Large Vision-Language Models via Causal Route Gating

Zhe Cheng [* 1]   Wenyu Chen [* 1]   Fode Zhang [1]   Dehuan Shen [2]

## Abstract

Large vision-language models (LVLMs) often hallucinate content that is fluent yet unsupported by the image, limiting their reliability in real-world deployment. We show that a key failure mode arises from route competition: even when visual tokens receive attention, the final token decision can be dominated by the textual pathway, causing the decoder to follow linguistic priors over visual evidence. To mitigate this, we propose a training-free, decision-aligned intervention that decomposes each attention head into a visual route and a text route, and estimates their token-level effects using an efficient one-forward/one-gradient approximation. These estimates reveal *route conflict* within heads and identify prior-dominant ones, enabling selective suppression of only the text route while keeping the visual route intact. Across five benchmarks spanning discriminative and generative settings, our method consistently reduces hallucination-related errors across models with limited impact on overall multimodal performance, while incurring a modest inference-time overhead.

## 1. Introduction

LVLMs have become a core interface for visual understanding, enabling systems that can answer questions about images and generate grounded descriptions (Yin et al., 2024a; Caffagni et al., 2024). However, a central obstacle to reliable deployment is hallucination: the model produces fluent, plausible content that is not supported by the image (Liu et al., 2024c). Importantly, this is not merely "poor writing"

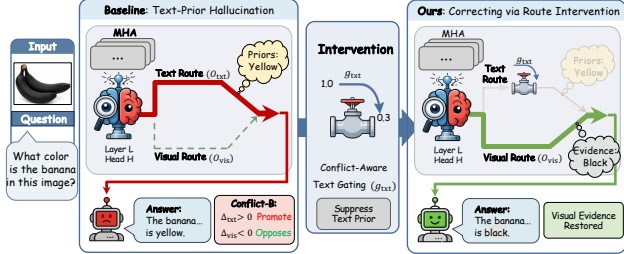

*Figure 1.* When language priors override visual inputs, the baseline model tends to produce hallucinated predictions (left). To address this, a conflict-aware intervention applies text-route gating to suppress language priors (middle), enabling the model to correctly rely on visual evidence after intervention (right).

or generic inaccuracy; rather, it often reflects a failure of grounding, where the model fills in missing visual details using language priors (Favero et al., 2024; Wu et al., 2025; Zhou et al., 2025). Such prior-driven fabrications directly undermine trustworthiness and can be harmful in safety-critical or decision-support settings (Wang et al., 2023b). At a high level, hallucination often reflects a mismatch between visual evidence and language priors: the model may attend to the image, yet the final token choice is still dominated by the textual pathway, favoring what is plausible over what is visible.

Training-free mitigation has become attractive for deployment. Among approaches that act at inference time without additional training, two representative paradigms emerge: (i) output-level decoding control, and (ii) proxy-guided internal intervention. The first paradigm modifies generation behavior through decoding-time heuristics or constraints (e.g., Visual Contrastive Decoding (VCD) (Leng et al., 2024), OPERA (Huang et al., 2024), MaskCD (Deng & Yang, 2025)). While effective, these methods primarily act as external policies and provide limited insight into which internal components drive prior-dominant errors. The second paradigm performs internal intervention by selecting modules using attention-based proxies (Chefer et al., 2021; Yüksekgönül et al., 2024), most notably VAR-style measures (Jiang et al., 2025; Zheng & Zhang, 2025; Jung et al., 2025) that quantify the attention share on visual tokens. Yet such proxies are inherently correlational: higher visual

[*]Equal contribution [1]Center of Statistical Research, School of Statistics and Data Science, Southwestern University of Finance and Economics, Chengdu, China. [2]Department of Biomedical Engineering, College of Design and Engineering, National University of Singapore, Singapore. Correspondence to: Fode Zhang <fredzh@swufe.edu.cn>.

*Proceedings of the 43rd International Conference on Machine Learning*, Seoul, South Korea. PMLR 306, 2026. Copyright 2026 by the author(s).

attention does not necessarily translate to larger causal contribution to the output (Jain & Wallace, 2019; Serrano & Smith, 2019), and softmax normalization can spuriously change the visual share by weakening textual competition (Vaswani et al., 2017). Moreover, proxy-based selection is typically paired with coarse head rescaling (Jiang et al., 2025; Yu et al., 2026), which inevitably suppresses both routes and may harm genuinely grounded reasoning.

To move beyond attention-based proxies, we expose a head-internal *visual route* and *text route*, so that we can intervene on either route at inference time (Fig. 1). For the current token $y^*$, we measure *decision-aligned* route effects using do-style interventions: $\Delta_{l,h}^{\text{vis}} = \ell_{l,h}(1,1) - \ell_{l,h}(0,1)$ and $\Delta_{l,h}^{\text{txt}} = \ell_{l,h}(1,1) - \ell_{l,h}(1,0)$. These effects directly reveal *route conflict*: a head is conflicting when $\text{sign}(\Delta_{l,h}^{\text{vis}}) \neq \text{sign}(\Delta_{l,h}^{\text{txt}})$, with Conflict-A $(+,-)$ and Conflict-B $(-,+)$. Within conflicting heads, we identify prior-dominant ones by $VRI_{l,h} = \frac{|\Delta_{l,h}^{\text{vis}}|}{|\Delta_{l,h}^{\text{vis}}|+|\Delta_{l,h}^{\text{txt}}|+\epsilon}$ (smaller $VRI_{l,h}$ indicates more text-driven behavior). At decoding time, we estimate $\Delta_{l,h}^{\text{vis}}$ and $\Delta_{l,h}^{\text{txt}}$ with a one-forward/one-gradient approximation, select Top-$k$ smallest $VRI_{l,h}$ *within the conflict sets*, and suppress only their text routes using a smooth rank-based schedule, with stronger suppression for Conflict-B than Conflict-A (visual routes remain intact). We evaluate our method on five benchmarks spanning discriminative and generative settings. Results indicate that the proposed method can reduce hallucination-related errors across models, with limited impact on overall multimodal performance.

**Contributions.** (1) We introduce route-level, decision-aligned causal effects for attention heads and a corresponding Visual Reliance Index. (2) We derive an efficient first-order estimator that enables per-step head selection during decoding. (3) We propose conflict-aware text-route gating that targets prior-dominant, decision-conflicting heads without suppressing visual routes. (4) We theoretically analyze VAR-style proxies, showing attention mass can be misaligned with decision-relevant visual grounding.

## 2. Related Work

**Hallucination in LVLMs.** Hallucination in LVLMs has received growing attention because it undermines visual grounding (Li et al., 2023; Wang et al., 2023a; Fu et al., 2025; Ye et al., 2023). Unlike textual LLMs, LVLMs exhibit prominent object hallucination (Rohrbach et al., 2018), where the model mentions objects or attributes not supported by the image. Mitigation methods span training-based alignment/fine-tuning (Liu et al., 2024a; Gunjal et al., 2024; Hu et al., 2023) and training-free inference-time control (Yin et al., 2024b; Tong et al., 2024; Chen et al., 2024; Zhang et al., 2025; Tang et al., 2025). On the training-free side, methods broadly fall into output-level decoding con-

trol and intervention-style approaches that act on internal computation at inference time (Sarkar et al., 2025). For example, PAI (Liu et al., 2024d) encourages stronger reliance on visual evidence, and latent-space steering methods such as VTI (Liu et al., 2025) modify hidden representations to suppress hallucination tendencies. Our work is most closely related to these training-free intervention paradigms, but differs in where and how we act: we compute token-level *within-head* route effects and apply route-specific gating at decoding time to reduce language-prior influence.

**Causal Interventions for Mechanistic Attribution.** Mechanistic interpretability (Elhage et al., 2021; Sharkey et al., 2025) increasingly emphasizes *interventional* attribution, where one perturbs internal states and measures the induced behavioral change (Geiger et al., 2021; 2025). Prior work on pruning and ablations suggests substantial redundancy in multi-head attention, but indiscriminate removals are often a blunt instrument and do not directly localize causal pathways (Michel et al., 2019; Voita et al., 2019). To pinpoint causal structure, Causal Mediation Analysis (CMA) (Vig et al., 2020a; Yang et al., 2025) and activation patching (a.k.a. causal tracing) (Zhang & Nanda, 2024) perform *targeted* interventions under controlled inputs to estimate component effects (Vig et al., 2020b; Geiger et al., 2021; Meng et al., 2022; Yang et al., 2025), and path-level methods (Qian et al., 2025) further attribute behaviors to specific information-flow paths (Goldowsky-Dill et al., 2023). Since head contributions can be highly interactive and non-modular, Causal Head Gating (CHG) learns differentiable head gates and searches for sufficient head sets directly from data, reducing reliance on hand-crafted prompt pairs (Nam et al., 2025). Building on this interventional line, we extend head-level interventions to *within-head* routes: unlike CHG's learned head gates, we estimate route-level effects to drive decoding-time route-specific gating that reduces language-prior dominance and maintains visual grounding.

## 3. Causal Route Effects

Figure 2 provides an overview; we now detail Step 1 (Calculate CRE) and later introduce conflict-aware gating.

### 3.1. Exact, Head-Internal Decomposition

We consider a pretrained LVLM parameterized by $f_\theta$. At a given decoding step, the model receives a prefix of length $L = M + T$, consisting of $M$ visual tokens indexed by $I_{\text{vis}} = \{1, \ldots, M\}$ and $T$ textual tokens indexed by $I_{\text{txt}} = \{M+1, \ldots, L\}$.

Many modern LVLMs employ Transformer-based language decoders with multi-head self-attention (MHA). At layer $l$, for attention head $h \in \{1, \ldots, H\}$, we use learned projections $W_{l,h}^Q, W_{l,h}^K, W_{l,h}^V \in \mathbb{R}^{d_{\text{model}} \times d_h}$, form-

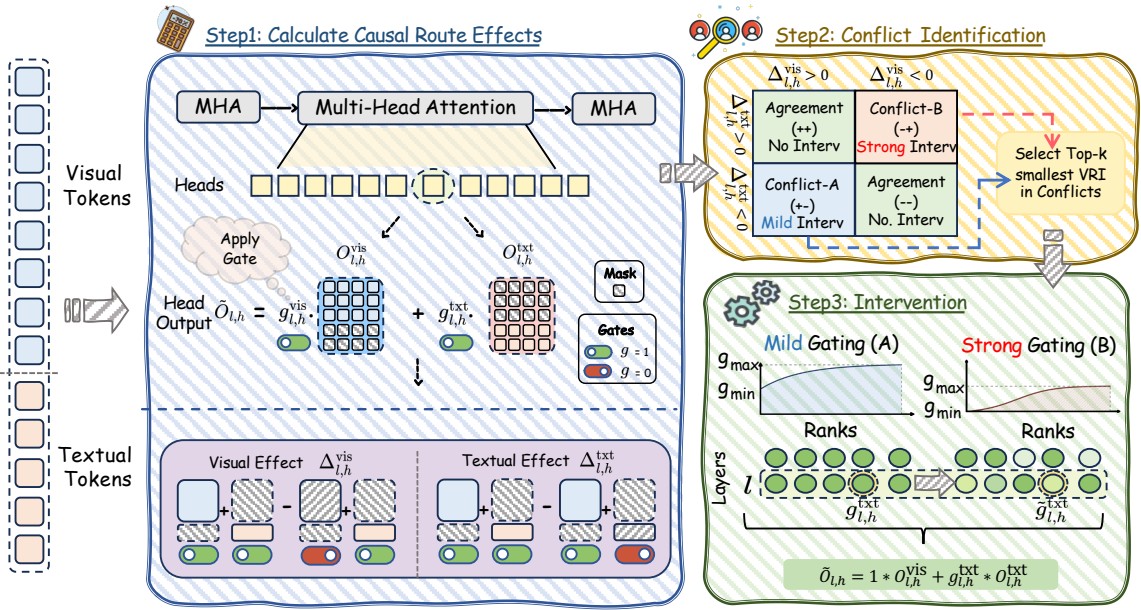

*Figure 2.* Overview of our CRG framework. Step 1 computes modal causal effects for each attention head by separating visual and textual components and measuring their contributions. Step 2 identifies conflicts by comparing the signs of visual and textual effects, distinguishing Agreement vs. Conflict (A/B), and selects the top-$k$ conflicting heads for intervention. Step 3 applies head gating, where Conflict-A heads receive mild gating and Conflict-B heads receive strong gating, reducing language-prior dominance while keeping image-grounded evidence.

ing $Q_{l,h}, K_{l,h}, V_{l,h}$ in the standard way. The causal attention weights are then computed by a masked softmax:

$$\alpha_{l,h} = \text{softmax}\left(\frac{Q_{l,h}K_{l,h}^{\top}}{\sqrt{d_h}} + M_{\text{causal}}\right),$$

where $M_{\text{causal}} \in \mathbb{R}^{L \times L}$ assigns $-\infty$ to future positions to enforce autoregressive attention. Finally, the head output is

$$O_{l,h} = \alpha_{l,h}V_{l,h}.$$

To isolate modality-specific contributions, we leverage the visual/textual index sets $I_{\text{vis}}$ and $I_{\text{txt}}$ introduced above and define diagonal selection matrices $S_{\text{vis}}, S_{\text{txt}} \in \{0,1\}^{L \times L}$ as

$$S_{\text{vis}} = \text{diag}\big(\mathbf{1}[1 \in I_{\text{vis}}], \ldots, \mathbf{1}[L \in I_{\text{vis}}]\big),$$
$$S_{\text{txt}} = \text{diag}\big(\mathbf{1}[1 \in I_{\text{txt}}], \ldots, \mathbf{1}[L \in I_{\text{txt}}]\big).$$

and they satisfy

$$S_{\text{vis}} + S_{\text{txt}} = I_L, \qquad S_{\text{vis}}S_{\text{txt}} = 0,$$

so the two masks are complementary and non-overlapping.

Applying these masks to the value matrix yields two route-specific value blocks, and consequently two route-specific head outputs:

$$O_{l,h}^{\text{vis}} = \alpha_{l,h}\big(S_{\text{vis}}V_{l,h}\big), \qquad O_{l,h}^{\text{txt}} = \alpha_{l,h}\big(S_{\text{txt}}V_{l,h}\big).$$

Since $V_{l,h} = (S_{\text{vis}} + S_{\text{txt}})V_{l,h}$, the head output splits exactly into visual and textual routes,

$$O_{l,h} = \alpha_{l,h}V_{l,h} = O_{l,h}^{\text{vis}} + O_{l,h}^{\text{txt}}.$$

**Propagation through the MHA output.** After computing all head outputs, the multi-head attention module concatenates them and applies the output projection:

$$Y_l^{\text{MHA}} = \text{Concat}_{h=1}^{H}(O_{l,h})\, W_l^O,$$
$$W_l^O \in \mathbb{R}^{Hd_h \times d_{\text{model}}}.$$

Because concatenation and the linear map $W_l^O$ are both linear operations, the same decomposition carries through the full MHA block:

$$Y_l^{\text{MHA}} = \text{Concat}_h(O_{l,h}^{\text{vis}})\, W_l^O \;+\; \text{Concat}_h(O_{l,h}^{\text{txt}})\, W_l^O$$
$$\triangleq\; Y_l^{\text{vis}} + Y_l^{\text{txt}}.$$

This gives an exact, layer-wise split of the MHA output into a visual-route component and a text-route component, which we will later exploit for route-specific interventions.

### 3.2. Interventional Gates and Causal Estimands

For each layer $l$ and head $h$, after computing the joint-softmax attention and forming the exact split $O_{l,h}^{\text{vis}}$ and $O_{l,h}^{\text{txt}}$, we attach head-internal, route-specific scalar gates

$g_{l,h}^{\text{vis}}, g_{l,h}^{\text{txt}} \in \mathbb{R}_{\geq 0}$ *before* the output projection $W_l^O$ (Qiu et al., 2025). The gated head output is

$$\tilde{O}_{l,h}(g_{l,h}^{\text{vis}}, g_{l,h}^{\text{txt}}) = g_{l,h}^{\text{vis}} O_{l,h}^{\text{vis}} + g_{l,h}^{\text{txt}} O_{l,h}^{\text{txt}}.$$

To quantify the effect of each route, we perform single-head interventions: we only modify the two route gates of one head $(l, h)$ and keep the rest of the network unchanged. As a reference, the baseline configuration sets all gates to one, i.e., $g_{l',h'}^{\text{vis}} = g_{l',h'}^{\text{txt}} = 1$ for every head $(l', h')$.

For the selected head $(l, h)$, we then consider binary gate values $g_{\text{vis}}, g_{\text{txt}} \in \{0, 1\}$: setting $g_{\text{vis}} = 0$ turns off the visual route of this head, and setting $g_{\text{txt}} = 0$ turns off its text route. All other heads remain at the baseline $(1, 1)$.

Let $f_{g_{\text{vis}}, g_{\text{txt}}}^{(l,h)}$ denote the intervened network in which head $(l, h)$ uses gates $(g_{\text{vis}}, g_{\text{txt}})$. Given a task-specific scalar decision score $\ell(\cdot)$, we define

$$\ell_{l,h}(g_{\text{vis}}, g_{\text{txt}}) := \ell\left(f_{g_{\text{vis}}, g_{\text{txt}}}^{(l,h)}\right).$$

Here $\ell_{l,h}(g_{\text{vis}}, g_{\text{txt}})$ is the decision score obtained under the single-head intervention on $(l, h)$ with gates $(g_{\text{vis}}, g_{\text{txt}})$. We define the *visual* and *text* Causal Route Effects (CRE) for head $(l, h)$ by the following do-effects:

$$\begin{aligned} \Delta_{l,h}^{\text{vis}} &:= \ell_{l,h}(1, 1) - \ell_{l,h}(0, 1), \\ \Delta_{l,h}^{\text{txt}} &:= \ell_{l,h}(1, 1) - \ell_{l,h}(1, 0). \end{aligned} \quad (1)$$

Intuitively, $\Delta_{l,h}^{\text{vis}}$ measures how much the decision score changes when we remove only the visual route of this head (while keeping its text route on), and $\Delta_{l,h}^{\text{txt}}$ is defined analogously. A positive $\Delta$ indicates that the corresponding route supports the decision (increases $\ell$), whereas a negative $\Delta$ indicates an inhibiting effect.

The choice of $\ell$ is task-dependent: it is a scalar score that summarizes the model's current decision, so that $\Delta_{l,h}^{\text{vis}}$ and $\Delta_{l,h}^{\text{txt}}$ quantify how the visual/text routes of head $(l, h)$ support or oppose that decision. For generative benchmarks, we set $\ell = \log p(y^*)$, the log-probability of the current token $y^*$, to measure how each route contributes to producing the desired output. For binary QA, we use the Yes/No margin $\ell = \log p(\text{Yes}) - \log p(\text{No})$, so that the sign of $\Delta$ directly indicates whether a route pushes the prediction toward Yes or toward No. All interventions are *local*: we only toggle the selected gates while keeping the rest of the computation fixed. Details of $\ell$ are provided in Appendix D.3.

To summarize whether a head's influence on the decision score $\ell$ is dominated by visual evidence or by textual context, we further define a normalized *Visual Reliance Index* (VRI):

$$\text{VRI}_{l,h} = \frac{|\Delta_{l,h}^{\text{vis}}|}{|\Delta_{l,h}^{\text{vis}}| + |\Delta_{l,h}^{\text{txt}}| + \varepsilon}. \quad (2)$$

where $\varepsilon > 0$ is a small stability constant for numerical stability (we use $\varepsilon = 10^{-8}$ in experiments). We use $\text{VRI}_{l,h}$ to rank heads for inference-time gating, and analogously define layer- or model-level indices by aggregation.

### 3.3. One-Forward Estimation

We estimate the binary do-effects using a *first-order* approximation that requires only *one forward* pass and a single *autograd gradient query* per decoded token. Specifically, we run a standard decode forward with all gates set to 1, caching the route-specific head outputs $O_{l,h}^{\text{vis}}$ and $O_{l,h}^{\text{txt}}$ (with the KV-cache intact). We then obtain the sensitivity $G_{l,h} = \partial \ell / \partial \tilde{O}_{l,h}$ by directly querying reverse-mode autodiff (e.g., via `torch.autograd.grad`) at the pre-$W_l^O$ tensor. This yields directional-derivative estimators along the two gate axes:

**Proposition 3.1** (Directional-derivative estimator). *At* $(g_{\text{vis}}, g_{\text{txt}}) = (1, 1)$,

$$\widehat{\Delta}_{l,h}^{\text{vis}} = \left\langle G_{l,h}, O_{l,h}^{\text{vis}} \right\rangle, \qquad \widehat{\Delta}_{l,h}^{\text{txt}} = \left\langle G_{l,h}, O_{l,h}^{\text{txt}} \right\rangle, \quad (3)$$

*which coincide with the first-order terms of the exact do-differences* $\Delta_{l,h}^{\text{vis}}$ *and* $\Delta_{l,h}^{\text{txt}}$. *(Proof in Appendix C.)*

*Remark* 3.2 (Coarse estimates for downstream intervention). Since $\widehat{\Delta}_{l,h}$ is first-order and local, we treat it as a coarse signal for the intervention in Sec. 4, which mainly depends on sign/ranking rather than precise effect magnitudes. This aligns with the broader insight that allocation-style objectives can often be optimized with coarse effect estimates (Casacuberta & Hardt, 2026).

To justify our first-order estimator, we **bound** its deviation from the exact do-effects under a Lipschitz condition on $G_{l,h}$ (Appendix B.1). We also validate it empirically by comparing $\hat{\Delta}$ with exact hard-intervention effects, with an emphasis on **sign consistency** that underlies our conflict taxonomy (Appendix B.2, B.3). We then compute the plug-in VRI as

$$\widehat{\text{VRI}}_{l,h} = \frac{|\widehat{\Delta}_{l,h}^{\text{vis}}|}{|\widehat{\Delta}_{l,h}^{\text{vis}}| + |\widehat{\Delta}_{l,h}^{\text{txt}}| + \varepsilon}.$$

## 4. Conflict-Aware Text-Route Intervention

Building on $(\hat{\Delta}_{l,h}^{\text{vis}}, \hat{\Delta}_{l,h}^{\text{txt}})$, we derive a lightweight, conflict-aware rule that selectively suppresses unreliable text contributions while preserving visual evidence (Fig. 2, Steps 2–3).

We categorize heads by the sign pattern of $(\hat{\Delta}_{l,h}^{\text{vis}}, \hat{\Delta}_{l,h}^{\text{txt}})$ into four regimes (Table 1). When the signs agree (++ or ——), both routes move $\ell$ in the same direction, so we leave these heads unchanged to avoid unnecessary perturbations. We intervene only under modality conflicts (+— or —+), and only through the text route $g_{l,h}^{\text{txt}}$ (keeping $g_{l,h}^{\text{vis}} = 1$).

*Table 1.* Sign taxonomy of route gains, where $\hat{\Delta} > 0$ increases the decision score $\ell$. We intervene only under conflicts (A/B) and leave agreement cases unchanged.

|  | $\hat{\Delta}^{\mathrm{vis}} > 0$ | $\hat{\Delta}^{\mathrm{vis}} < 0$ |
|---|---|---|
| $\hat{\Delta}^{\mathrm{txt}} > 0$ | **Agreement** $(+,+)$ | **Conflict-B** $(-,+)$ |
| $\hat{\Delta}^{\mathrm{txt}} < 0$ | **Conflict-A** $(+,-)$ | **Agreement** $(-,-)$ |

---

**Algorithm 1** Conflict-Aware Text-Route Intervention

---

**Require:** Per-head estimates $\hat{\Delta}_{l,h}^{\mathrm{vis}}, \hat{\Delta}_{l,h}^{\mathrm{txt}}$; VRI scores $v_{l,h}$;
    top-$k$ values $k$; gate ranges $(g_{\min}^A, g_{\max}^A), (g_{\min}^B, g_{\max}^B)$;
    $\gamma, \epsilon$.

**Ensure:** Text gates $\{g_{l,h}^{\mathrm{txt}}\}$ (default 1).

1: Build conflict sets $\mathcal{H}_A, \mathcal{H}_B$ in Eq. (4).
2: **for** $\mathcal{H} \in \{\mathcal{H}_A, \mathcal{H}_B\}$ **do**
3:     Select $\mathcal{S} = \mathrm{TopKSmallest}(\{v_{l,h}\}, k)$ within $\mathcal{H}$.
4:     Sort $\mathcal{S}$ by $v_{l,h}$ ascending; assign rank $i$.
5:     Set $s_i = i/(|\mathcal{S}| - 1)$ (if $|\mathcal{S}| > 1$); else set $g^{\mathrm{txt}} = g_{\min}$.
6:     Assign $g_{(i)}^{\mathrm{txt}} = g_{\min} + (g_{\max} - g_{\min}) \cdot \mathrm{clip}(s_i^\gamma, \epsilon, 1-\epsilon)$.
7:     Update the corresponding heads' $g_{l,h}^{\mathrm{txt}} \leftarrow g_{(i)}^{\mathrm{txt}}$.
8: **end for**
9: **return** $\{g_{l,h}^{\mathrm{txt}}\}$.

---

**Conflict-A** $(+,-)$**: visual increases $\ell$ while text decreases $\ell$.** Here the visual route promotes the current decision score, but the text route acts as an obstruction. This pattern is consistent with text-side noise or local linguistic mismatch rather than a reliable correction to visual evidence. Our goal is *stabilization*: we mildly suppress the text route on a small subset of heads so that visual evidence can dominate without distorting generation.

**Conflict-B** $(-,+)$**: visual decreases $\ell$ while text increases $\ell$ (hallucination hallmark).** Here visual evidence discourages the current decision, yet the text route actively promotes it. This corresponds to the regime where language priors override vision. Our goal is *prior cutting*: we strongly suppress the text route (possibly near-zero) on the most prior-dominant heads, so that visual counter-evidence can be reflected in the logits/decision score.

### 4.1. VRI-Based Intervention

Algorithm 1 summarizes the procedure.

**Selecting the most relevant heads using VRI.** Recall the two modality-conflict regimes: Conflict-A $(+,-)$ and Conflict-B $(-,+)$. We denote their head sets by

$$\mathcal{H}_A = \{(l,h) : \hat{\Delta}_{l,h}^{\mathrm{vis}} > 0, \hat{\Delta}_{l,h}^{\mathrm{txt}} < 0\},$$
$$\mathcal{H}_B = \{(l,h) : \hat{\Delta}_{l,h}^{\mathrm{vis}} < 0, \hat{\Delta}_{l,h}^{\mathrm{txt}} > 0\}. \quad (4)$$

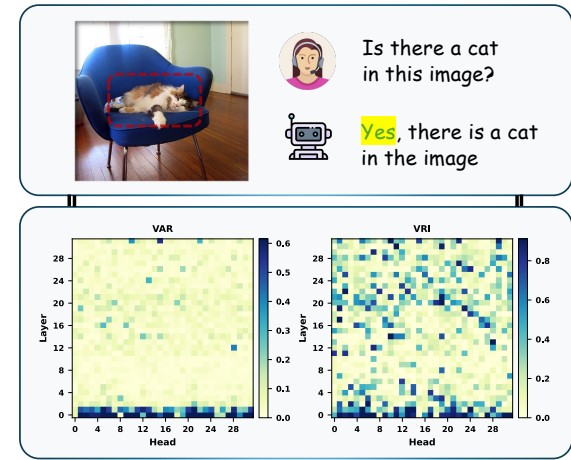

*Figure 3.* Per-layer, per-head VAR (left) and VRI (right) for the generated token "Yes" from LLaVA-1.5-7B. VAR reflects visual attention allocation, while VRI reflects relative visual reliance computed from decision-aligned route effects.

Within each conflict set $\mathcal{H} \in \{\mathcal{H}_A, \mathcal{H}_B\}$, we assign each head a scalar score $v_{l,h} = \mathrm{VRI}_{l,h}$. We then select a small subset $\mathcal{S} \subseteq \mathcal{H}$ of size $k$ by taking the heads with the smallest VRI values: $\mathcal{S} = \mathrm{TopKSmallest}(\{v_{l,h} : (l,h) \in \mathcal{H}\}, k)$. This concentrates the intervention on a limited number of heads, improving controllability and reducing unintended side effects. More broadly, this top-$k$ selection can be viewed as a budgeted allocation step, where coarse effect-derived scores may suffice for near-optimal allocation value (Casacuberta & Hardt, 2026).

**Rank-range gate schedule.** Given the selected heads $\mathcal{S}$ with $|\mathcal{S}| = n$, we sort them by VRI in ascending order and assign rank $i \in \{0, \dots, n-1\}$. For $n > 1$, we define $s_i = i/(n-1)$ and map it to a gate value: $g_{(i)}^{\mathrm{txt}} = g_{\min} + (g_{\max} - g_{\min}) \cdot \mathrm{clip}(s_i^\gamma, \epsilon, 1-\epsilon)$, where $\gamma > 0$ controls the nonlinearity of the rank-to-gate mapping and $\epsilon > 0$ avoids extreme values. Only heads in $\mathcal{S}$ receive scheduled gates, while all other heads keep $g_{l,h}^{\mathrm{txt}} = 1$. This schedule ensures a smooth and monotonic suppression strength across the selected heads instead of a hard threshold.

**Conflict-specific gate ranges.** We use different gate ranges for the two conflicts: (i) Conflict-A applies *mild* text suppression with $(g_{\min}^A, g_{\max}^A) = (0.5, 1.0)$, treating the text route as potentially noisy but not adversarial. (ii) Conflict-B applies *strong* text suppression with $(g_{\min}^B, g_{\max}^B) = (0, 0.5)$, aiming to cut the prior-supporting pathway when language priors promote $y^*$ despite visual opposition. In both cases, $g_{l,h}^{\mathrm{vis}}$ remains 1. We keep these ranges fixed across models as a simple mild/strong semantic split, while model- and task-level adaptation comes from VRI-based head selection and within-range rank ordering.

*Table 2.* Main results on POPE tasks. The best results are bolded.

| Setting | Method | LLaVA-1.5-7B | | Qwen-VL-Chat | | Qwen2.5-VL-7B-Instruct | |
|---|---|---|---|---|---|---|---|
| | | Accuracy↑ | F1-Score↑ | Accuracy↑ | F1-Score↑ | Accuracy↑ | F1-Score↑ |
| Random | Regular | 83.29 | 81.33 | 84.63 | 82.61 | 84.52 | 84.62 |
| | VCD | 87.73 | 87.16 | 86.93 | 85.46 | 87.82 | 88.23 |
| | OPERA | 89.20 | 88.81 | 85.71 | 84.64 | 89.72 | **90.02** |
| | PAI | 86.33 | 84.56 | 85.38 | 85.54 | 89.54 | 88.72 |
| | VTI | 89.50 | 88.89 | 86.73 | 85.59 | 90.43 | 89.44 |
| | **CRG(ours)** | **90.30** (+7.01) | **89.51** (+8.18) | **89.46** (+4.83) | **88.33** (+5.72) | **91.45** (+6.93) | 89.23 (+4.61) |
| Popular | Regular | 81.88 | 80.06 | 83.63 | 81.53 | 84.67 | 85.13 |
| | VCD | 85.38 | 85.06 | 85.17 | 83.68 | 86.89 | 87.35 |
| | OPERA | 86.64 | 86.62 | 84.82 | 83.99 | 87.57 | 88.12 |
| | PAI | 85.33 | 83.62 | 84.20 | 83.10 | 88.14 | 87.32 |
| | VTI | 87.36 | **86.69** | 85.67 | 84.48 | 89.23 | 86.23 |
| | **CRG(ours)** | **88.40** (+6.52) | 86.54 (+6.48) | **87.63** (+4.00) | **86.55** (+5.02) | **90.73** (+6.06) | **89.34** (+4.21) |
| Adversarial | Regular | 78.96 | 77.57 | 81.03 | 79.30 | 82.79 | 83.15 |
| | VCD | 80.88 | 81.33 | 83.10 | 82.04 | 84.78 | 84.63 |
| | OPERA | 81.24 | 81.38 | 82.67 | 79.89 | 84.93 | 85.17 |
| | PAI | 83.17 | 81.67 | 82.19 | 82.06 | 85.12 | 84.77 |
| | VTI | 82.57 | 82.11 | 83.13 | 82.16 | 85.78 | 85.14 |
| | **CRG(ours)** | **84.43** (+5.47) | **83.77** (+6.20) | **84.70** (+3.67) | **83.78** (+4.48) | **86.98** (+4.19) | **87.07** (+3.92) |

## 4.2. Why a High Visual Attention Ratio Does Not Guarantee Grounded Outputs

A popular proxy for "visual reliance" is the *Visual Attention Ratio* (VAR) (Jiang et al., 2025), which measures how much attention mass a head assigns to visual keys. Using the same visual index set $I_{\text{vis}}$ as Sec. 3, for a decoding query position $i$ we adopt $\text{VAR}_{l,h} := \sum_{j \in I_{\text{vis}}} \alpha_{ij}^{(l,h)}$, optionally averaged over query positions. However, VAR is not decision-aligned. We use our decision-aligned visual route effect $\widehat{\Delta}_{l,h}^{\text{vis}}$ to expose this misalignment, and use VRI to summarize normalized modality reliance. In Fig. 3, VAR and VRI share a similar pattern in the first few layers, where many heads allocate substantial attention to visual keys. However, in subsequent middle layers VAR becomes nearly flat even for the correct "Yes" token, while VRI still exhibits nontrivial structure. This indicates that, for this QA pattern, attention mass alone is not a reliable proxy for decision-relevant visual grounding.

The reason is that VAR depends only on attention weights, while the visual route effect depends on both weights and value content. Recall our first-order estimator in Eq. (3), $\widehat{\Delta}_{l,h}^{\text{vis}} \approx \langle G_{l,h}, O_{l,h}^{\text{vis}} \rangle$, where $G_{l,h} := \partial\ell / \partial\tilde{O}_{l,h}$ for a task-dependent scalar score $\ell$. Writing $O_{l,h}^{\text{vis}} = \sum_{j \in I_{\text{vis}}} \alpha_{ij}^{(l,h)} v_j^{(l,h)}$ yields [1] $\widehat{\Delta}_{l,h}^{\text{vis}} \approx \sum_{j \in I_{\text{vis}}} \alpha_{ij}^{(l,h)} \langle G_{l,h}, v_j^{(l,h)} \rangle$. Thus, even when $\text{VAR}_{l,h}$ is large, the signed projections $\langle G_{l,h}, v_j^{(l,h)} \rangle$ can be small, can-

---

[1] For vectors $a, b_j$ and scalars $c_j$, $\langle a, \sum_j c_j b_j \rangle = \sum_j \langle a, c_j b_j \rangle = \sum_j c_j \langle a, b_j \rangle$.

cel out, or be negative, making the net visual influence on the decision weak or even harmful—information that VAR discards.

Moreover, VAR is confounded by softmax competition. Let $\alpha_j = \exp(s_j) / \sum_k \exp(s_k)$, where $s_j$ is the pre-softmax attention logit, and $R = \sum_{j \in I_{\text{vis}}} \alpha_j$ (the per-query VAR). For a textual key $k \in I_{\text{txt}}$, $\frac{\partial R}{\partial s_k} = -R\alpha_k < 0$, so decreasing textual logits increases $R$ even if all visual logits stay unchanged. Therefore, a higher VAR may reflect weakened textual competition rather than stronger visual grounding.

In summary, VAR is neither sufficient nor necessary for identifying decision-critical (or hallucination-prone) components, motivating our use of decision-aligned CRE in Eq. (1). For further analysis and experiment comparison, readers can refer to Appendix A.

## 5. Experiments

**Models.** We adopt LLaVA-1.5-7B (Liu et al., 2024b), Qwen-VL-Chat-7B (Bai et al., 2023), and Qwen2.5-VL-7B-Instruct (Bai et al., 2025). Appendix E.1 further reports results on more advanced LVLMs.

**Benchmarks and Metrics.** We evaluate our method on **five** benchmarks spanning discriminative and generative settings. We use POPE (Li et al., 2023) for binary hallucination QA (Accuracy/F1), CHAIR (Rohrbach et al., 2018) (caption-level object hallucination), MME (Fu et al., 2025) for comprehensive multimodal evaluation (overall score), MMHal-Bench (Sun et al., 2024) for open-ended halluci-

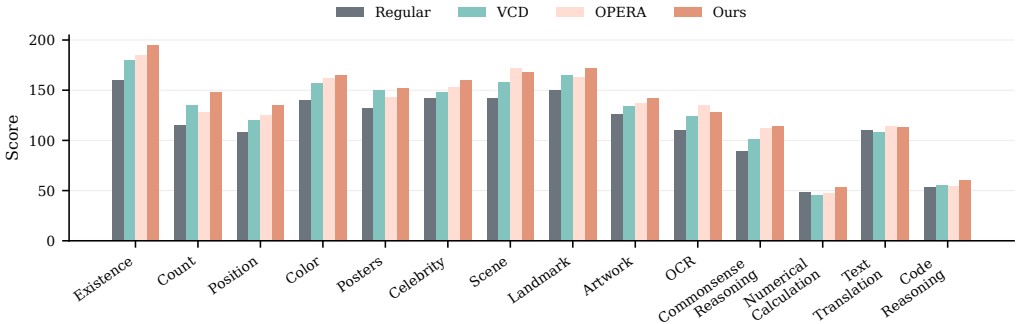

*Figure 4.* Category-wise MME scores (higher is better) comparing Regular decoding with VCD, OPERA, and our method.

nation assessment (official hallucination metrics), and AM-BER (Wang et al., 2023a), a unified benchmark covering both generative and discriminative hallucination behaviors. For detailed Benchmarks and Metrics, readers can refer to Appendix D.1.

**Baselines and Implementation Details.** We use greedy decoding unless otherwise specified. We denote our inference-time method as **Causal Route Gating (CRG)**, and compare against representative training-free baselines, including VCD (Leng et al., 2024), OPERA (Huang et al., 2024), PAI (Liu et al., 2024d), and VTI (Liu et al., 2025). Detailed experimental settings, model configurations, and decoding hyperparameters are provided in Appendix D.2. Additional analyses and extensions are reported in Appendix E.1 (stronger LVLMs), Appendix E.2 (combining with VCD), and Appendix A.4 (comparison with VAR-style proxies).

### 5.1. Main Results

**Results on POPE** As shown in Table 2, our method achieves the best accuracy and is competitive on F1 across settings. Relative to Regular, our method improves performance by **6.3%** accuracy and **7.0%** F1 on LLaVA-1.5-7B, **4.2%** accuracy and **5.1%** F1 on Qwen-VL-Chat, and **5.7%** accuracy and **4.2%** F1 on Qwen2.5-VL-7B-Instruct. Compared with the strongest baseline VTI, our method still yields consistent gains, improving accuracy/F1 by **1.2%/0.7%** on LLaVA-1.5-7B, **2.1%/2.1%** on Qwen-VL-Chat, and **1.2%/1.6%** on Qwen2.5-VL-7B-Instruct.

**Results on MME.** As shown in Fig. 4, our method consistently improves over Regular decoding across almost all categories, with clear gains on grounding-sensitive skills such as *Existence*, *Count*, *Position*, and *Color*, while also improving higher-level categories (e.g., *Commonsense Reasoning* and *Numerical Calculation*). These results indicate that our conflict-aware suppression reduces hallucination without sacrificing general multi-modal performance.

**Results on CHAIR.** As shown in Table 3, our method

*Table 3.* Results on CHAIR benchmark. Max new tokens are set to be 512.

| Method | LLaVA-1.5-7B | | | | Qwen-VL-Chat | | | |
|---|---|---|---|---|---|---|---|---|
| | $C_S \downarrow$ | $C_I \downarrow$ | Recall↑ | Len | $C_S \downarrow$ | $C_I \downarrow$ | Recall↑ | Len |
| Regular | 52.8 | 15.9 | 77.3 | 93.4 | 50.2 | 14.2 | 76.4 | 92.1 |
| VCD | 51.6 | 14.9 | 77.2 | 101.9 | 47.3 | 13.7 | 75.3 | 98.1 |
| OPERA | 44.6 | 12.8 | 78.5 | 95.3 | 42.3 | 12.3 | 79.4 | 97.5 |
| PAI | 39.1 | 12.4 | 76.9 | 94.4 | 37.2 | 11.9 | **82.3** | 96.3 |
| VTI | 37.6 | 12.9 | **79.3** | 93.8 | 35.6 | 11.5 | 81.2 | 95.3 |
| CRG | **34.2** | **11.2** | 77.8 | 98.1 | **32.4** | **10.4** | 81.6 | 96.6 |

*Table 4.* Results on AMBER under the official evaluation pipeline (LLaVA-1.5-7B). AMBER Score is computed as $(100 - \text{CHAIR} + F1)/2$.

| Method | CHAIR↓ | Accuracy↑ | F1↑ | AMBER Score↑ |
|---|---|---|---|---|
| Regular | 8.3 | 72.7 | 73.7 | 82.70 |
| CRG | **4.6** (-3.7) | **77.4** (+4.7) | **77.5** (+3.8) | **86.45** (+3.75) |

achieves the lowest hallucination rates on both back-bones: on LLaVA-1.5-7B, $C_S/C_I$ drop from $52.8/15.9$ to $34.2/11.2$ while keeping recall close to Regular (77.8 vs. 77.3); on Qwen-VL-Chat, $C_S/C_I$ decrease from $50.2/14.2$ to $32.4/10.4$ with recall improving relative to Regular (81.6 vs. 76.4). These improvements come with only a modest increase in length, indicating reduced hallucination without aggressively shortening the generation.

**Results on MMHal-Bench.** Figure 5 provides a category-wise breakdown, where our method consistently dominates the radar plots across most MMHal categories for both LLaVA-1.5-7B and Qwen-VL-Chat. For detailed results of MMHal-Bench, please refer to Table 16 in Appendix E.3.

**Results on AMBER.** Table 4 shows that our method reduces hallucinations (CHAIR: $8.3 \rightarrow 4.6$) and improves discriminative performance (F1: $73.7 \rightarrow 77.5$) on LLaVA-1.5-7B. As a result, the AMBER Score increases by +3.75. Detailed AMBER results are provided in Table 17 (Appendix E.3).

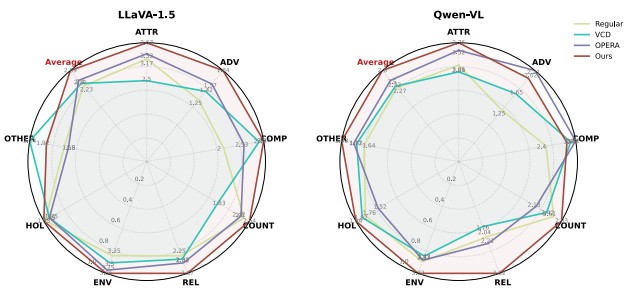

*Figure 5.* Per-category radar comparison on MMHal-Bench for two VLMs (LLaVA-1.5-7B and Qwen-VL-Chat).

*Table 5.* Ablation study on LLaVA-1.5-7B across four benchmarks.

| Method | POPE-Avg↑ | CHAIR$C_S$↓ | MMHal↑ | MME↑ |
|---|---|---|---|---|
| Regular | 81.37 | 52.8 | 2.23 | 1640 |
| CRG (A+B) | **87.71** | **34.2** | **2.69** | **1897.42** |
| w/o A | 87.47 | 34.7 | 2.54 | 1892.18 |
| w/o B | 87.31 | 35.2 | 2.62 | 1884.32 |

### 5.2. Ablation Study and Hyperparameter Analysis

**Conflict-Aware Strategy Selection (A/B).** To identify which conflict intervention contributes most, we ablate our strategy by removing one component at a time while keeping the rest unchanged. On LLaVA-1.5-7B, we evaluate POPE, CHAIR, MMHal, and MME, comparing our full strategy (A+B) with w/o A (only Conflict B) and w/o B (only Conflict A). Table 5 shows that A+B consistently outperforms the LLaVA-1.5-7B baseline on all four benchmarks. Both ablations also improve over the baseline, but fall short of the full method, indicating that Conflict A and Conflict B provide complementary signals. Moreover, w/o A performs slightly better than w/o B in most cases, suggesting Conflict B tends to be the stronger single component in this setting.

**Intervention Layer Range.** We ablate the intervention layer range ($L_{start}, L_{end}$) on POPE-`popular` for LLaVA-1.5-7B and Qwen-VL-Chat-7B by sweeping $L_{start} \in [5, 24]$ and $L_{end} \in [L_{start} + 7, 32]$. Fig. 6 shows accuracy heatmaps (star: best). LLaVA-1.5-7B peaks at $(8, 19)$ with 0.8840, which is consistent with the trend in Fig. 3, while Qwen-VL-Chat-7B peaks at $(10, 24)$ with 0.8763. Both models favor intervening on a contiguous mid-to-upper layer block; very shallow or very late ranges are consistently less effective.

**Hyperparameter Sensitivity of $k$ and $\gamma$.** Figure 7 reports CHAIR results when varying the head budget $k$ and the rank-to-gate nonlinearity $\gamma$ (lower $C_S/C_I$ is better; higher F1 is better). **Varying $k$** (left), increasing $k$ generally reduces $C_S$ and $C_I$ and improves F1, with the best performance at a moderate budget (around $k \approx 11$); larger $k$ brings diminishing returns and slightly lowers F1. **Varying $\gamma$** (right), moderate values yield the best trade-off, with peak F1 and lowest hallucination rates around $\gamma \approx 0.3$–$0.7$ (best near

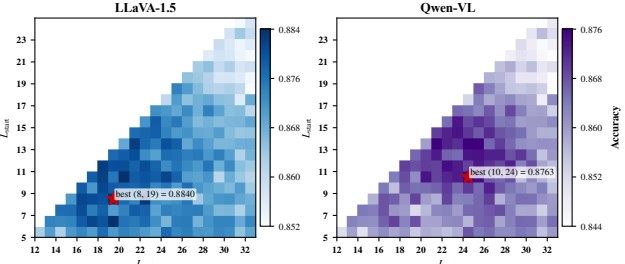

*Figure 6.* Layer-range ablation for intervention (POPE Setting Popular). Heatmap shows accuracy for intervening on layers $[L_{start}, L_{end}]$; the star marks the best range.

*Table 6.* Runtime and peak GPU memory overhead under the same decoding setup.

| Method | Avg. Latency↓ | GPU Memory↓ |
|---|---|---|
| Regular | 61.3 ms (×1.00) | 14945 MB (×1.00) |
| VCD | **123.2 ms** (×2.01) | 15749 MB (×1.05) |
| OPERA | 435.7 ms (×7.12) | 22706 MB (×1.52) |
| **CRG** | 139.2 ms (×2.27) | **14950 MB** (×1.00) |

0.5), while extreme $\gamma$ values are less effective.

### 5.3. Complexity Analysis

**Runtime and memory overhead.** Table 6 reports average latency and peak GPU memory under the same decoding setup. CRG incurs essentially no additional memory cost (×1.00), since we backpropagate only through lightweight gate scalars with frozen model parameters. Notably, this overhead is close to VCD (2.01×; 123.2 ms), while OPERA is substantially slower (7.12×; 435.7 ms). Latency is reported per decoding step under the same setup using PyTorch eager attention. Additional cost analysis is provided in Appendix G.

## 6. Conclusion and Limitations

**Conclusion.** Hallucination in LVLMs is often driven by a mismatch between visual evidence and linguistic priors, where plausible continuations override what is supported by the image. We move beyond attention-based proxies

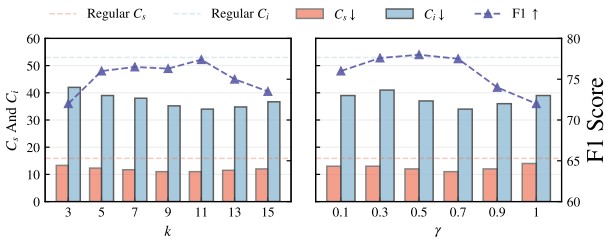

*Figure 7.* Hyperparameter sensitivity of $k$ and $\gamma$.

by exposing two head-internal routes—a visual route and a text route—and measuring their decision-aligned effects on the current token via do-style interventions. This route-level view reveals *route conflict* and enables a conflict-aware inference-time intervention that selectively suppresses only the prior-dominant text route while keeping the visual route intact, avoiding the indiscriminate suppression induced by coarse head rescaling. Across five benchmarks spanning discriminative and generative settings, our method consistently reduces hallucination-related errors across models with limited impact on overall multimodal performance, with a modest and controllable inference-time overhead.

**Limitations.** First, route effects are estimated using a lightweight one-forward/one-gradient approximation; the estimate is local to the current decoding step and may be noisy under ambiguous evidence or strong cross-layer interactions. More broadly, CRG is targeted at conflict-mediated hallucinations, and is not expected to correct cases where both visual and textual routes share the same bias, or errors dominated by perception, OCR, or multi-step reasoning failures. Second, the overhead grows with generation length since effect estimation and gating are applied during decoding; while the per-token cost is modest in our experiments, long-form generation remains more expensive. Third, the method requires white-box access to internal activations/gradients and inference-time patching, which may not apply to closed-source APIs or restricted deployment settings. Finally, selectively suppressing the text route can make the model more conservative, potentially reducing descriptive richness when visual evidence is weak or when the task legitimately benefits from linguistic priors; balancing faithfulness and helpfulness remains an important direction for future work. We discuss these scope and deployment boundaries in more detail in Appendix F.

## Acknowledgements

We thank the anonymous reviewers for their constructive comments. This work was supported in part by the Sichuan Science and Technology Program under Grant 2024ZYD0135, in part by the National Natural Science Foundation of China under Grants 12071372 and 12201395, and in part by the Fundamental Research Funds for the Central Universities (JBK2507005).

## Impact Statement

This paper proposes **CRG**, an inference-time intervention that estimates token-level, decision-aligned effects of within-head visual and text routes and selectively suppresses prior-dominant text routes to improve visual grounding and reduce hallucinated claims in vision–language models. We use VRI as a lightweight ranking signal to focus interventions on a small subset of heads. The primary societal benefit is improved reliability and interpretability of model behavior in downstream applications. Potential risks include misuse to produce more convincing misinformation, uneven effects across demographic or cultural contexts, and over-reliance on automated mitigation despite residual failure modes. We therefore recommend careful evaluation across diverse settings, transparent reporting of limitations, and human oversight for high-stakes deployment.

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

# Appendix

## Contents

# A. Analysis of VAR-Style Attention Proxies

## A.1. CRE vs. VAR-Style Proxies

The main text argues that attention-mass proxies, such as VAR (Jiang et al., 2025), are not decision-aligned for diagnosing and intervening on hallucination behaviors in LVLMs. In our framework, the target of interest is the decision score $\ell$ (defined in the main text for discriminative and generative settings), and we quantify how each head-route causally changes $\ell$ via do-interventions. Concretely, for head $(l, h)$, we define the visual-route and textual-route CRE as $\Delta_{l,h}^{\mathrm{vis}} := \ell_{l,h}(1,1) - \ell_{l,h}(0,1)$ and $\Delta_{l,h}^{\mathrm{txt}} := \ell_{l,h}(1,1) - \ell_{l,h}(1,0)$, where $\ell_{l,h}(g_{\mathrm{vis}}, g_{\mathrm{txt}})$ denotes the score under route gates $(g_{\mathrm{vis}}, g_{\mathrm{txt}}) \in \{0,1\}^2$. We further normalize the relative reliance via the $\mathrm{VRI}_{l,h} := \frac{|\Delta_{l,h}^{\mathrm{vis}}|}{|\Delta_{l,h}^{\mathrm{vis}}| + |\Delta_{l,h}^{\mathrm{txt}}| + \varepsilon}$, where $\varepsilon > 0$ is a small constant for numerical stability.

In contrast, VAR summarizes only the amount of attention mass assigned to visual tokens. Let $I_{\mathrm{vis}}$ be the index set of visual tokens, and let $\alpha_{ij}^{(l,h)}$ denote the attention weight from query position $i$ to key position $j$ at head $(l, h)$. A typical VAR definition takes the form $\mathrm{VAR}_{l,h}(i) := \sum_{j \in I_{\mathrm{vis}}} \alpha_{ij}^{(l,h)}$, optionally averaged over query positions $i$ (e.g., over decoded positions or over all tokens in the prefix). Crucially, $\mathrm{VAR}_{l,h}$ does not condition on (i) the content carried by value vectors, nor (ii) how such content aligns with the local sensitivity of the score $\ell$. Therefore, VAR is inherently a correlational proxy: it measures where the head attends, but not whether the attended information supports (or opposes) the current decision.

## A.2. Theoretical Analysis: What VAR Can (and Cannot) Guarantee

We formalize the limitation of VAR by connecting it to a first-order, decision-aligned approximation of route effects. Let $v_j^{(l,h)}$ be the value vector at key position $j$ for head $(l, h)$, and define the visual-route output as $O_{l,h}^{\mathrm{vis}}(i) := \sum_{j \in I_{\mathrm{vis}}} \alpha_{ij}^{(l,h)} v_j^{(l,h)}$. Let $G_{l,h}(i)$ denote the local sensitivity (gradient-like) vector used in the main text, so that the first-order approximation of the visual-route effect can be written as $\widehat{\Delta}_{l,h}^{\mathrm{vis}}(i) := \langle G_{l,h}(i), O_{l,h}^{\mathrm{vis}}(i) \rangle$. Expanding the definition yields $\widehat{\Delta}_{l,h}^{\mathrm{vis}}(i) = \sum_{j \in I_{\mathrm{vis}}} \alpha_{ij}^{(l,h)} \langle G_{l,h}(i), v_j^{(l,h)} \rangle$, which makes explicit that the decision-aligned quantity depends not only on attention weights $\alpha$, but also on the signed alignment terms $\langle G, v \rangle$.

**Lemma A.1** (VAR provides only a coarse upper bound). *Define $m_{l,h}(i) := \max_{j \in I_{\mathrm{vis}}} |\langle G_{l,h}(i), v_j^{(l,h)} \rangle|$. Then the first-order visual-route effect satisfies*

$$\left| \widehat{\Delta}_{l,h}^{\mathrm{vis}}(i) \right| \leq m_{l,h}(i) \cdot \mathrm{VAR}_{l,h}(i).$$

*Proof.* Recall that the first-order visual-route effect can be written as a weighted sum over visual keys:

$$\widehat{\Delta}_{l,h}^{\mathrm{vis}}(i) = \sum_{j \in I_{\mathrm{vis}}} \alpha_{ij}^{(l,h)} \langle G_{l,h}(i), v_j^{(l,h)} \rangle.$$

Taking the absolute value and applying the triangle inequality yields

$$\left| \widehat{\Delta}_{l,h}^{\mathrm{vis}}(i) \right| = \left| \sum_{j \in I_{\mathrm{vis}}} \alpha_{ij}^{(l,h)} \langle G_{l,h}(i), v_j^{(l,h)} \rangle \right| \leq \sum_{j \in I_{\mathrm{vis}}} \alpha_{ij}^{(l,h)} \left| \langle G_{l,h}(i), v_j^{(l,h)} \rangle \right|.$$

By the definition of $m_{l,h}(i)$, for every $j \in I_{\mathrm{vis}}$ we have

$$\left| \langle G_{l,h}(i), v_j^{(l,h)} \rangle \right| \leq m_{l,h}(i).$$

Substituting this bound into the previous inequality gives

$$\sum_{j \in I_{\mathrm{vis}}} \alpha_{ij}^{(l,h)} \left| \langle G_{l,h}(i), v_j^{(l,h)} \rangle \right| \leq m_{l,h}(i) \sum_{j \in I_{\mathrm{vis}}} \alpha_{ij}^{(l,h)}.$$

Finally, noting that $\sum_{j \in I_{\mathrm{vis}}} \alpha_{ij}^{(l,h)} = \mathrm{VAR}_{l,h}(i)$ by definition, we obtain

$$\left| \widehat{\Delta}_{l,h}^{\mathrm{vis}}(i) \right| \leq m_{l,h}(i) \cdot \mathrm{VAR}_{l,h}(i),$$

which completes the proof. $\square$

**Implication.** Lemma A.1 clarifies that VAR can at best constrain the magnitude of a decision-aligned effect up to an unknown, head- and context-dependent factor $m_{l,h}(i)$. However, $m_{l,h}(i)$ depends on value content and its alignment with the score sensitivity, which is not observable from attention weights alone. Consequently, ranking or selecting heads solely by $\text{VAR}_{l,h}$ is not theoretically justified as a decision-aligned criterion, and can be unreliable especially when $\langle G, v \rangle$ varies widely or changes sign across tokens.

**Lemma A.2** (A tighter second-moment upper bound). *For $j \in I_{\text{vis}}$, define the (signed) alignment $a_j := \langle G_{l,h}(i), v_j^{(l,h)} \rangle$. When $\text{VAR}_{l,h}(i) > 0$, define the normalized visual attention weights $\tilde{\alpha}_{ij}^{(l,h)} := \frac{\alpha_{ij}^{(l,h)}}{\text{VAR}_{l,h}(i)}, j \in I_{\text{vis}}$, so that $\sum_{j \in I_{\text{vis}}} \tilde{\alpha}_{ij}^{(l,h)} = 1$. Define the (attention-weighted) second-moment factor $s_{l,h}(i) := \left( \sum_{j \in I_{\text{vis}}} \tilde{\alpha}_{ij}^{(l,h)} a_j^2 \right)^{1/2}$. Then the first-order visual-route effect satisfies*

$$\left| \widehat{\Delta}_{l,h}^{\text{vis}}(i) \right| \leq \text{VAR}_{l,h}(i) \cdot s_{l,h}(i) \leq \text{VAR}_{l,h}(i) \cdot m_{l,h}(i),$$

*where $m_{l,h}(i) := \max_{j \in I_{\text{vis}}} |a_j|$ as in Lemma A.1.*

*Proof.* Starting from the definition of the first-order effect,

$$\widehat{\Delta}_{l,h}^{\text{vis}}(i) = \sum_{j \in I_{\text{vis}}} \alpha_{ij}^{(l,h)} a_j = \text{VAR}_{l,h}(i) \sum_{j \in I_{\text{vis}}} \tilde{\alpha}_{ij}^{(l,h)} a_j.$$

Taking absolute values yields

$$\left| \widehat{\Delta}_{l,h}^{\text{vis}}(i) \right| = \text{VAR}_{l,h}(i) \left| \sum_{j \in I_{\text{vis}}} \tilde{\alpha}_{ij}^{(l,h)} a_j \right|.$$

Applying the Cauchy–Schwarz inequality to the weighted sum gives

$$\left| \sum_{j \in I_{\text{vis}}} \tilde{\alpha}_{ij}^{(l,h)} a_j \right| \leq \left( \sum_{j \in I_{\text{vis}}} \tilde{\alpha}_{ij}^{(l,h)} \right)^{1/2} \left( \sum_{j \in I_{\text{vis}}} \tilde{\alpha}_{ij}^{(l,h)} a_j^2 \right)^{1/2}.$$

Since $\sum_{j \in I_{\text{vis}}} \tilde{\alpha}_{ij}^{(l,h)} = 1$, the first factor equals 1, and we obtain

$$\left| \sum_{j \in I_{\text{vis}}} \tilde{\alpha}_{ij}^{(l,h)} a_j \right| \leq \left( \sum_{j \in I_{\text{vis}}} \tilde{\alpha}_{ij}^{(l,h)} a_j^2 \right)^{1/2} = s_{l,h}(i).$$

Therefore,

$$\left| \widehat{\Delta}_{l,h}^{\text{vis}}(i) \right| \leq \text{VAR}_{l,h}(i) \cdot s_{l,h}(i).$$

Finally, because $|a_j| \leq m_{l,h}(i)$ for all $j \in I_{\text{vis}}$, we have $a_j^2 \leq m_{l,h}(i)^2$, and hence

$$s_{l,h}(i)^2 = \sum_{j \in I_{\text{vis}}} \tilde{\alpha}_{ij}^{(l,h)} a_j^2 \leq m_{l,h}(i)^2 \sum_{j \in I_{\text{vis}}} \tilde{\alpha}_{ij}^{(l,h)} = m_{l,h}(i)^2,$$

$\square$

*Remark* A.3. Lemma A.2 sharpens Lemma A.1 by replacing the worst-case factor $m_{l,h}(i)$ with a data-dependent RMS factor $s_{l,h}(i)$, which can be much smaller when only a few attended visual tokens have large alignment magnitude.

**Proposition A.4** (When VAR becomes informative: sign-consistent alignment). *Assume $\text{VAR}_{l,h}(i) > 0$. For $j \in I_{\text{vis}}$, let $a_j := \langle G_{l,h}(i), v_j^{(l,h)} \rangle$. Suppose that $\{a_j\}_{j \in I_{\text{vis}}}$ is sign-consistent, i.e., either $a_j \geq 0$ for all $j \in I_{\text{vis}}$ or $a_j \leq 0$ for all $j \in I_{\text{vis}}$. Moreover, assume that there exist scalars $\underline{a}_{l,h}(i) \leq \overline{a}_{l,h}(i)$ such that $a_j \in \left[ \underline{a}_{l,h}(i), \overline{a}_{l,h}(i) \right]$, for all $j \in I_{\text{vis}}$.*

*Then the first-order visual-route effect obeys the two-sided bound*

$$\underline{a}_{l,h}(i) \, \text{VAR}_{l,h}(i) \leq \widehat{\Delta}_{l,h}^{\text{vis}}(i) \leq \overline{a}_{l,h}(i) \, \text{VAR}_{l,h}(i).$$

*Proof.* By definition, when $\mathrm{VAR}_{l,h}(i) > 0$ the normalized visual attention weights

$$\tilde{\alpha}_{ij}^{(l,h)} := \frac{\alpha_{ij}^{(l,h)}}{\mathrm{VAR}_{l,h}(i)}, \qquad j \in I_{\mathrm{vis}},$$

satisfy $\sum_{j \in I_{\mathrm{vis}}} \tilde{\alpha}_{ij}^{(l,h)} = 1$ and $\tilde{\alpha}_{ij}^{(l,h)} \geq 0$. Using the expansion of the first-order effect,

$$\widehat{\Delta}_{l,h}^{\mathrm{vis}}(i) = \sum_{j \in I_{\mathrm{vis}}} \alpha_{ij}^{(l,h)} a_j = \mathrm{VAR}_{l,h}(i) \sum_{j \in I_{\mathrm{vis}}} \tilde{\alpha}_{ij}^{(l,h)} a_j.$$

The quantity $\sum_{j \in I_{\mathrm{vis}}} \tilde{\alpha}_{ij}^{(l,h)} a_j$ is a convex combination of $\{a_j\}_{j \in I_{\mathrm{vis}}}$. Since each $a_j \in [\underline{a}_{l,h}(i), \overline{a}_{l,h}(i)]$, any convex combination also lies in the same interval, namely

$$\underline{a}_{l,h}(i) \leq \sum_{j \in I_{\mathrm{vis}}} \tilde{\alpha}_{ij}^{(l,h)} a_j \leq \overline{a}_{l,h}(i).$$

Multiplying both sides by the positive scalar $\mathrm{VAR}_{l,h}(i)$ yields

$$\underline{a}_{l,h}(i) \,\mathrm{VAR}_{l,h}(i) \leq \widehat{\Delta}_{l,h}^{\mathrm{vis}}(i) \leq \overline{a}_{l,h}(i) \,\mathrm{VAR}_{l,h}(i),$$

which completes the proof. $\qquad\square$

*Remark* A.5. This proposition formalizes a *sufficient condition* under which VAR may correlate with decision-aligned effect: the attended visual values must be roughly aligned with the score sensitivity (no sign flips or severe cancellations). Outside this regime, large VAR does not imply large $\widehat{\Delta}^{\mathrm{vis}}$ (cf. Appendix A.3).

**Theorem A.6** (Non-identifiability of decision-aligned influence from attention alone). *Fix a head $(l, h)$ and a query position $i$. Assume that the attention weights $\{\alpha_{ij}^{(l,h)}\}_j$ are fixed (equivalently, the queries and keys are fixed), and let*

$$\mathrm{VAR}_{l,h}(i) := \sum_{j \in I_{\mathrm{vis}}} \alpha_{ij}^{(l,h)}.$$

*Let $G_{l,h}(i)$ be any nonzero sensitivity vector with $\|G_{l,h}(i)\|_2 > 0$. Suppose the value vectors are norm-bounded as $\|v_j^{(l,h)}\|_2 \leq V$ for all $j \in I_{\mathrm{vis}}$, where $V > 0$. Then there exist two feasible value assignments $\{v_j^{(l,h)}\}_{j \in I_{\mathrm{vis}}}$ and $\{v_j'^{(l,h)}\}_{j \in I_{\mathrm{vis}}}$ that yield the same attention weights $\alpha_{ij}^{(l,h)}$ but produce opposite first-order visual-route effects:*

$$\widehat{\Delta}_{l,h}^{\mathrm{vis}}(i) = +V\|G_{l,h}(i)\|_2 \,\mathrm{VAR}_{l,h}(i), \qquad \widehat{\Delta}_{l,h}^{\mathrm{vis}\prime}(i) = -V\|G_{l,h}(i)\|_2 \,\mathrm{VAR}_{l,h}(i).$$

*Consequently, any proxy that depends only on attention allocation—including attention mass ratios such as VAR—cannot, in general, identify the* sign *of the decision-aligned visual influence.*

*Proof.* Let $u := \frac{G_{l,h}(i)}{\|G_{l,h}(i)\|_2}$, so that $\|u\|_2 = 1$. Define two value assignments on visual keys by setting, for all $j \in I_{\mathrm{vis}}$,

$$v_j^{(l,h)} := Vu, \qquad v_j'^{(l,h)} := -Vu.$$

Both assignments satisfy the norm constraint $\|v_j^{(l,h)}\|_2 = \|v_j'^{(l,h)}\|_2 = V$. Moreover,

$$\langle G_{l,h}(i),\, v_j^{(l,h)} \rangle = \langle G_{l,h}(i),\, Vu \rangle = V\|G_{l,h}(i)\|_2,$$

$$\langle G_{l,h}(i),\, v_j'^{(l,h)} \rangle = -V\|G_{l,h}(i)\|_2.$$

Using the definition of the first-order visual-route effect,

$$\begin{aligned}
\widehat{\Delta}_{l,h}^{\mathrm{vis}}(i) &= \sum_{j \in I_{\mathrm{vis}}} \alpha_{ij}^{(l,h)} \langle G_{l,h}(i),\, v_j^{(l,h)} \rangle \\
&= V\|G_{l,h}(i)\|_2 \sum_{j \in I_{\mathrm{vis}}} \alpha_{ij}^{(l,h)} \\
&= V\|G_{l,h}(i)\|_2 \,\mathrm{VAR}_{l,h}(i),
\end{aligned}$$

Similarly,

$$\widehat{\Delta}_{l,h}^{\mathrm{vis}\prime}(i) = \sum_{j\in I_{\mathrm{vis}}} \alpha_{ij}^{(l,h)} \langle G_{l,h}(i),\, v_j^{\prime(l,h)}\rangle$$

$$= -V\|G_{l,h}(i)\|_2\,\mathrm{VAR}_{l,h}(i).$$

Since both constructions keep $\alpha_{ij}^{(l,h)}$ unchanged while flipping the sign of the induced effect, the sign of $\widehat{\Delta}_{l,h}^{\mathrm{vis}}(i)$ is not identifiable from attention allocation alone. $\qquad\square$

**Corollary A.7.** *Even if two heads have identical* $\mathrm{VAR}_{l,h}(i)$, *their decision-aligned visual effects can have opposite signs and differ by* $2V\|G_{l,h}(i)\|_2\,\mathrm{VAR}_{l,h}(i)$. *Hence, selecting heads based solely on VAR admits arbitrarily bad mistakes under adversarial (but norm-bounded) value alignments.*

### A.3. Three Canonical Failure Modes of VAR

We present three minimal constructions showing that large (or small) VAR does not imply large (or small) decision-aligned visual influence. For simplicity, we fix a head $(l, h)$ and a query position $i$, and consider visual keys $j \in I_{\mathrm{vis}}$ only.

1. **Orthogonality (large VAR, zero effect).** Assume $\mathrm{VAR}_{l,h}(i) = 1$, i.e., all attention mass is assigned to visual tokens, but $\langle G_{l,h}(i), v_j^{(l,h)}\rangle = 0$ for every $j \in I_{\mathrm{vis}}$. Then $\widehat{\Delta}_{l,h}^{\mathrm{vis}}(i) = \sum_{j\in I_{\mathrm{vis}}} \alpha_{ij}^{(l,h)} \cdot 0 = 0$. Thus, a head may appear "highly visual" under VAR while contributing no visual evidence to the current decision.

2. **Cancellation (large VAR, small effect by sign mixing).** Consider two visual tokens $j_1, j_2 \in I_{\mathrm{vis}}$ with $\alpha_{ij_1}^{(l,h)} = \alpha_{ij_2}^{(l,h)} = 1/2$, so $\mathrm{VAR}_{l,h}(i) = 1$. Let $\langle G_{l,h}(i), v_{j_1}^{(l,h)}\rangle = +1$ and $\langle G_{l,h}(i), v_{j_2}^{(l,h)}\rangle = -1$. Then $\widehat{\Delta}_{l,h}^{\mathrm{vis}}(i) = \frac{1}{2}(+1) + \frac{1}{2}(-1) = 0$. Hence, even when VAR is maximal, the net decision-aligned effect can vanish due to signed cancellations.

3. **Harmful visual content (large VAR, negative effect).** VAR is nonnegative by construction and cannot encode whether the attended visual content supports or contradicts the target token. Suppose $\mathrm{VAR}_{l,h}(i)$ is large, but $\langle G_{l,h}(i), v_j^{(l,h)}\rangle < 0$ for the dominant attended visual keys $j$. Then $\widehat{\Delta}_{l,h}^{\mathrm{vis}}(i)$ becomes negative, meaning the visual route *reduces* the score $\ell$ for the current token (i.e., it provides counter-evidence). Such "harmful" visual influence is particularly relevant under route conflicts, yet it is invisible to VAR-based diagnostics.

*Table 7.* A summary of canonical (counterexample) regimes showing that attention-mass proxies such as VAR are not decision-aligned. Here $a_j = \langle G_{l,h}(i), v_j^{(l,h)}\rangle$, $\tilde{\alpha}_{ij} = \alpha_{ij}/\mathrm{VAR}_{l,h}(i)$, and $\mu := \sum_{j\in I_{\mathrm{vis}}} \tilde{\alpha}_{ij} a_j$ so that $\widehat{\Delta}_{l,h}^{\mathrm{vis}}(i) = \mathrm{VAR}_{l,h}(i) \cdot \mu$.

| | **High VAR** ($\mathrm{VAR}_{l,h}(i) \approx 1$) | **Low VAR** ($\mathrm{VAR}_{l,h}(i) \approx 0$) |
|---|---|---|
| **Near-zero alignment** ($\mu \approx 0$) | *Orthogonality:* $a_j = 0\ \forall j$ 
 *Cancellation:* sign-mixed $a_j$ 
 $\Rightarrow \widehat{\Delta}^{\mathrm{vis}} \approx 0$ even if VAR is large | Trivial small upper bound: 
 $|\widehat{\Delta}^{\mathrm{vis}}| \leq \mathrm{VAR} \cdot m$ 
 $\Rightarrow$ small effect is possible but uninformative |
| **Negative alignment** ($\mu < 0$) | *Harmful visual content:* dominant $a_j < 0$ 
 $\Rightarrow \widehat{\Delta}^{\mathrm{vis}} < 0$ despite high VAR 
 (VAR cannot reveal the sign) | Less common but possible: 
 small VAR yet negative influence 
 (rare, bounded by $\mathrm{VAR} \cdot m$) |

Table 7 summarizes the three canonical failure modes. Taken together, these constructions show that VAR characterizes only attention allocation, while decision-aligned route effects depend on content-sensitive alignment. This motivates using CRE/VRI as the primary signals for diagnosing and selecting intervention targets, rather than relying on attention-mass proxies.

### A.4. Empirical Alignment of VAR with Decision-Aligned Effects

**Setup.** On POPE-popular, we compute at the answer-critical decoding step the attention-mass proxy $\mathrm{VAR}_{l,h}(i) = \sum_{j\in I_{\mathrm{vis}}} \alpha_{ij}^{(l,h)}$, the decision-aligned first-order visual effect $\widehat{\Delta}_{l,h}^{\mathrm{vis}}(i)$ (from our one-forward/one-gradient estimate), and the relative reliance $\widehat{\mathrm{VRI}}_{l,h}(i) = \frac{|\widehat{\Delta}_{l,h}^{\mathrm{vis}}(i)|}{|\widehat{\Delta}_{l,h}^{\mathrm{vis}}(i)| + |\widehat{\Delta}_{l,h}^{\mathrm{txt}}(i)| + \varepsilon}$.

**Metric.** We report Spearman rank correlations over all head–example pairs.

Table 8. Spearman rank correlations between VAR and decision-aligned quantities at the answer-critical token.

| | $\rho_1 = \text{corr}_S(\text{VAR}, |\widehat{\Delta}^{\text{vis}}|)$ | $\rho_2 = \text{corr}_S(\text{VAR}, \widehat{\text{VRI}}_{l,h})$ |
|---|---|---|
| POPE (popular) | 0.5766 | 0.5300 |

**Interpretation.** VAR exhibits a moderate monotonic association with decision-aligned magnitudes on this split. However, this does *not* make VAR a decision-aligned diagnostic: VAR cannot identify the *sign* of visual influence, nor detect cancellation or harmful visual evidence (Appendix A.3), and thus cannot support reliable head selection under route conflicts. Our theoretical results show that attention allocation alone is insufficient for identifying decision-aligned influence in general (Theorem A.6), motivating CRE/VRI as primary signals for intervention.

**Empirical evidence for the "harmful visual content" regime.** Consistent with the third failure mode in Appendix A.3 (high VAR but negative alignment), on POPE-popular at the answer-critical token, the visual-route effect is negative in about half of the head–example pairs: $\Pr(\widehat{\Delta}^{\text{vis}} < 0) = 0.503$. Importantly, this does not vanish among heads with the largest visual attention mass: conditioning on the top 10% VAR heads (VAR $\geq 0.140$, the 90th percentile), we still have $\Pr(\widehat{\Delta}^{\text{vis}} < 0 \mid \text{VAR in top 10\%}) = 0.515$. Therefore, even when a head "looks" highly visual under VAR, its visual route can frequently provide counter-evidence (i.e., reduce the decision score), which is invisible to attention-mass proxies.

Table 9. Negative visual effects remain common even among high-VAR heads (POPE-popular).

| Split | $\Pr(\widehat{\Delta}^{\text{vis}} < 0)$ | $\Pr(\widehat{\Delta}^{\text{vis}} < 0 \mid \text{VAR top 10\%})$ | 90th pct. VAR |
|---|---|---|---|
| popular | 0.503 | 0.515 | 0.140 |

# B. Validity of the First-Order Do-Effect Approximation

We provide both theoretical and empirical evidence for the validity of our one-forward/one-gradient first-order approximation to head-level do-effects: (i) we derive a deterministic error bound under a mild Lipschitz-type regularity condition along the gating path, (ii) we give a margin-based sufficient condition for sign reliability, and (iii) we perform an exact sanity check on a small subset by comparing against controlled single-head interventions.

## B.1. First-Order Error Bound

**Proposition B.1** (First-order error bound). *Assume the map* $\tau \mapsto G_{l,h}(\tau, 1)$ *is L-Lipschitz on* $[0, 1]$, *i.e.,*

$$\|G_{l,h}(\tau_1, 1) - G_{l,h}(\tau_2, 1)\|_F \leq L|\tau_1 - \tau_2|, \quad \forall \tau_1, \tau_2 \in [0, 1].$$

*Then*

$$\left|\Delta_{l,h}^{\text{vis}} - \widehat{\Delta}_{l,h}^{\text{vis}}\right| \leq \frac{L}{2} \left\|O_{l,h}^{\text{vis}}\right\|_F,$$

*and symmetrically for the text route.*

*Proof.* From **Proposition 3.1** and the path-integral representation,

$$\Delta_{l,h}^{\text{vis}} - \widehat{\Delta}_{l,h}^{\text{vis}} = \int_0^1 \left\langle G_{l,h}(\tau, 1) - G_{l,h}(1, 1), O_{l,h}^{\text{vis}} \right\rangle d\tau.$$

Take absolute values and apply Cauchy–Schwarz and triangle inequality:

$$\left|\Delta_{l,h}^{\text{vis}} - \widehat{\Delta}_{l,h}^{\text{vis}}\right| \leq \|O_{l,h}^{\text{vis}}\|_F \int_0^1 \left\|G_{l,h}(\tau, 1) - G_{l,h}(1, 1)\right\|_F d\tau.$$

Under the Lipschitz condition $\|G_{l,h}(\tau, 1) - G_{l,h}(1, 1)\|_F \leq L(1 - \tau)$,

$$\int_0^1 \left\|G_{l,h}(\tau, 1) - G_{l,h}(1, 1)\right\|_F d\tau \leq \int_0^1 L(1 - \tau) \, d\tau = \frac{L}{2}.$$

Combining yields the stated bound. The proof for the textual route follows by symmetry. $\square$

*Remark* B.2. In practice, we observe $\|G_{l,h}(\tau, 1) - G_{l,h}(1, 1)\|_F$ grows approximately linearly as $\tau$ moves away from 1 for most heads, supporting the bounded-rate assumption.

### B.2. Sign Reliability

Proposition B.1 provides a deterministic bound on the approximation residual $R_{l,h}^{\mathrm{vis}} \triangleq \Delta_{l,h}^{\mathrm{vis}} - \widehat{\Delta}_{l,h}^{\mathrm{vis}}$. This immediately yields a sufficient condition under which the *sign* of the estimated do-effect is guaranteed to be correct.

**Lemma B.3** (Sign reliability under bounded error). *If* $\left|\Delta_{l,h}^{\mathrm{vis}} - \widehat{\Delta}_{l,h}^{\mathrm{vis}}\right| \leq \epsilon_{l,h}^{\mathrm{vis}}$, *then*

$$\widehat{\Delta}_{l,h}^{\mathrm{vis}} > \epsilon_{l,h}^{\mathrm{vis}} \Rightarrow \Delta_{l,h}^{\mathrm{vis}} > 0, \qquad \widehat{\Delta}_{l,h}^{\mathrm{vis}} < -\epsilon_{l,h}^{\mathrm{vis}} \Rightarrow \Delta_{l,h}^{\mathrm{vis}} < 0.$$

*In particular, whenever* $\left|\widehat{\Delta}_{l,h}^{\mathrm{vis}}\right| > \epsilon_{l,h}^{\mathrm{vis}}$, *we have* $\mathrm{sign}(\Delta_{l,h}^{\mathrm{vis}}) = \mathrm{sign}(\widehat{\Delta}_{l,h}^{\mathrm{vis}})$.

*Proof.* Assume the estimator error is bounded by

$$\left|\Delta_{l,h}^{\mathrm{vis}} - \widehat{\Delta}_{l,h}^{\mathrm{vis}}\right| \leq \epsilon_{l,h}^{\mathrm{vis}}.$$

If $\widehat{\Delta}_{l,h}^{\mathrm{vis}} > \epsilon_{l,h}^{\mathrm{vis}}$, then

$$\begin{aligned}
\Delta_{l,h}^{\mathrm{vis}} &= \widehat{\Delta}_{l,h}^{\mathrm{vis}} + \left(\Delta_{l,h}^{\mathrm{vis}} - \widehat{\Delta}_{l,h}^{\mathrm{vis}}\right) \\
&\geq \widehat{\Delta}_{l,h}^{\mathrm{vis}} - \left|\Delta_{l,h}^{\mathrm{vis}} - \widehat{\Delta}_{l,h}^{\mathrm{vis}}\right| \\
&\geq \widehat{\Delta}_{l,h}^{\mathrm{vis}} - \epsilon_{l,h}^{\mathrm{vis}} \\
&> 0.
\end{aligned}$$

Similarly, if $\widehat{\Delta}_{l,h}^{\mathrm{vis}} < -\epsilon_{l,h}^{\mathrm{vis}}$, then

$$\begin{aligned}
\Delta_{l,h}^{\mathrm{vis}} &= \widehat{\Delta}_{l,h}^{\mathrm{vis}} + \left(\Delta_{l,h}^{\mathrm{vis}} - \widehat{\Delta}_{l,h}^{\mathrm{vis}}\right) \\
&\leq \widehat{\Delta}_{l,h}^{\mathrm{vis}} + \left|\Delta_{l,h}^{\mathrm{vis}} - \widehat{\Delta}_{l,h}^{\mathrm{vis}}\right| \\
&\leq \widehat{\Delta}_{l,h}^{\mathrm{vis}} + \epsilon_{l,h}^{\mathrm{vis}} \\
&< 0.
\end{aligned}$$

$\square$

**Corollary B.4** (Instantiating Lemma B.3 via Proposition B.1). *Under the conditions of Proposition B.1, define*

$$\epsilon_{l,h}^{\mathrm{vis}} = \frac{L}{2} \left\|O_{l,h}^{\mathrm{vis}}\right\|_F, \qquad \epsilon_{l,h}^{\mathrm{txt}} = \frac{L}{2} \left\|O_{l,h}^{\mathrm{txt}}\right\|_F.$$

*Then, if* $\left|\widehat{\Delta}_{l,h}^{\mathrm{vis}}\right| > \epsilon_{l,h}^{\mathrm{vis}}$, *the sign of* $\widehat{\Delta}_{l,h}^{\mathrm{vis}}$ *matches that of* $\Delta_{l,h}^{\mathrm{vis}}$; *and symmetrically, if* $\left|\widehat{\Delta}_{l,h}^{\mathrm{txt}}\right| > \epsilon_{l,h}^{\mathrm{txt}}$, *the sign of* $\widehat{\Delta}_{l,h}^{\mathrm{txt}}$ *matches that of* $\Delta_{l,h}^{\mathrm{txt}}$.

**Implication for sign regimes.** If both margins hold simultaneously, $\left|\widehat{\Delta}_{l,h}^{\mathrm{vis}}\right| > \epsilon_{l,h}^{\mathrm{vis}}$ and $\left|\widehat{\Delta}_{l,h}^{\mathrm{txt}}\right| > \epsilon_{l,h}^{\mathrm{txt}}$, then the sign pattern of $\left(\widehat{\Delta}_{l,h}^{\mathrm{vis}}, \widehat{\Delta}_{l,h}^{\mathrm{txt}}\right)$ coincides with that of $\left(\Delta_{l,h}^{\mathrm{vis}}, \Delta_{l,h}^{\mathrm{txt}}\right)$, so the head's regime $(++, +-, -+, --)$ is certified.

### B.3. Exact Validation on a Small Subset

While Proposition B.1 bounds the first-order remainder under a smoothness condition, we further provide an *exact* sanity check on a small subset to empirically verify that the proposed first-order estimator faithfully captures both the *sign* and the *relative ranking* of head-level do-effects.

**Protocol.** For each example, we consider a fixed decoding step with a fixed prefix (image, question, and generated prefix tokens) and evaluate the effect on a scalar objective $\ell = \log p(\text{Yes}) - \log p(\text{No})$ at that step (the same $\ell$ used throughout our method). We first run a *baseline* single-step decode forward with all gates set to 1, obtaining $\ell_0 \triangleq \ell(1,1)$ and caching the route-specific head outputs $O_{l,h}^{\text{vis}}$ and $O_{l,h}^{\text{txt}}$ (with the KV-cache intact). We then compute the first-order estimates $\widehat{\Delta}_{l,h}^{\text{vis}}$ and $\widehat{\Delta}_{l,h}^{\text{txt}}$ at $(g_{\text{vis}}, g_{\text{txt}}) = (1,1)$ using the local sensitivity $G_{l,h} = \partial \ell / \partial \tilde{O}_{l,h}$ and the inner products in Proposition B.1.

To obtain the *exact* do-differences, we re-run the same *single-step* decode forward while intervening on *only one* head-route at a time: for the visual route, we set $g_{l,h}^{\text{vis}} = 0$ and keep all other gates (including $g_{l,h}^{\text{txt}}$) at 1, yielding $\ell_{l,h}^{\text{vis}-\text{off}}$; for the text route, we set $g_{l,h}^{\text{txt}} = 0$ and keep all others at 1, yielding $\ell_{l,h}^{\text{txt}-\text{off}}$. We define the exact do-differences as

$$\Delta_{l,h}^{\text{vis}} \triangleq \ell_0 - \ell_{l,h}^{\text{vis}-\text{off}}, \qquad \Delta_{l,h}^{\text{txt}} \triangleq \ell_0 - \ell_{l,h}^{\text{txt}-\text{off}}.$$

Importantly, all exact evaluations reuse the same prefix and the same KV-cache so that each $\Delta$ reflects a controlled, local intervention at the same decoding step (rather than differences induced by divergent generation trajectories).

**Subset selection.** We perform this validation on a small subset of $N = 50$ examples (POPE). For each example, we evaluate exact do-differences on $K = 32$ head-route pairs, consisting of (i) the $\text{TopSmallest} - K/2$ heads ranked by $\widehat{\text{VRI}}$ (most relevant to our gating) and (ii) $K/2$ uniformly sampled heads (to cover the long tail). This keeps the cost manageable while directly probing the heads that our method is likely to intervene on.

**Metrics.** We report three complementary metrics: (i) *Pearson/Spearman correlation* between $\widehat{\Delta}$ and $\Delta$ across evaluated head-route pairs, capturing whether the first-order estimator preserves relative magnitudes and rankings;

*Table 10.* Correlations between the first-order estimates $\widehat{\Delta}$ and exact do-differences $\Delta$ on a small validation subset. Correlation is higher on large-magnitude heads (Top) that dominate gating decisions.

| Route | Subset | #Pairs | Pearson $r$ | Spearman $\rho$ |
|-------|--------|--------|-------------|-----------------|
| Vis | All (Top + Random) | 1600 | 0.90 | 0.88 |
| Txt | All (Top + Random) | 1600 | 0.86 | 0.83 |
| Vis | TopSmallest-16 by $\widehat{\text{VRI}}$ | 800 | 0.95 | 0.93 |
| Vis | Random-16 | 800 | 0.78 | 0.74 |
| Txt | TopSmallest-16 by $\widehat{\text{VRI}}$ | 800 | 0.92 | 0.90 |
| Txt | Random-16 | 800 | 0.70 | 0.66 |

(ii) *sign agreement* $\mathbb{I}[\text{sign}(\widehat{\Delta}) = \text{sign}(\Delta)]$, which directly validates the reliability of our sign-based regime taxonomy.

*Table 11.* Sign agreement between $\widehat{\Delta}$ and exact $\Delta$ on a small validation subset. The last row reports the fraction of head-step pairs whose *two-route* sign pattern is simultaneously correct, directly supporting the regime taxonomy based on $(\text{sign}(\Delta^{\text{vis}}), \text{sign}(\Delta^{\text{txt}}))$.

| Route | #Pairs | Sign agreement (%) |
|-------|--------|--------------------|
| Vis | 1600 | 93.1 |
| Txt | 1600 | 90.4 |
| Both routes correct (vis $\wedge$ txt) | 1600 | 86.7 |

**Takeaway.** Table 10 and Table 11 show that the first-order estimates $\widehat{\Delta}$ closely track the exact do-differences $\Delta$ on a held-out subset. They preserve both *ranking* (high Pearson/Spearman correlation) and *sign* (over 90% agreement), with the strongest alignment on large-magnitude heads selected for gating. Moreover, the two-route sign pattern is simultaneously

correct for most head-step pairs, directly supporting our sign-based regime taxonomy. Overall, these results validate $\widehat{\Delta}$ as an accurate and efficient proxy for head-level do-effects in our intervention pipeline.

## C. Proofs of Main Results

For readability, we collect proofs of the main-text proposition in Appendix C. Proofs of results introduced only in the appendix are provided **inline** within the corresponding appendix sections. We further justify the validity of our first-order estimates, both theoretically and empirically, in Appendix B.

**Proposition 3.1 [Directional-derivative estimator]** *At* $(g_{\text{vis}}, g_{\text{txt}}) = (1, 1)$,

$$\widehat{\Delta}_{l,h}^{\text{vis}} = \left\langle G_{l,h}(1,1), O_{l,h}^{\text{vis}} \right\rangle, \qquad \widehat{\Delta}_{l,h}^{\text{txt}} = \left\langle G_{l,h}(1,1), O_{l,h}^{\text{txt}} \right\rangle,$$

*Moreover,* $\widehat{\Delta}_{l,h}^{\text{vis}} = \frac{\partial \ell_{l,h}}{\partial g_{\text{vis}}}(1,1)$ *and* $\widehat{\Delta}_{l,h}^{\text{txt}} = \frac{\partial \ell_{l,h}}{\partial g_{\text{txt}}}(1,1)$. *The exact do-differences admit the path-integral form, and the approximation error is characterized in Proposition B.1.*

*Proof.* Fix a layer–head pair $(l, h)$. Recall that the route decomposition is computed once from the baseline run at $(g_{\text{vis}}, g_{\text{txt}}) = (1, 1)$ and then *kept fixed* for all gate values. That is, $O_{l,h}^{\text{vis}}$ and $O_{l,h}^{\text{txt}}$ do not depend on $(g_{\text{vis}}, g_{\text{txt}})$; only their linear combination is gated. By definition of the gated head output

$$\tilde{O}_{l,h}(g_{\text{vis}}, g_{\text{txt}}) = g_{\text{vis}} O_{l,h}^{\text{vis}} + g_{\text{txt}} O_{l,h}^{\text{txt}},$$

the route activations $O_{l,h}^{\text{vis}}, O_{l,h}^{\text{txt}}$ are constants with respect to $(g_{\text{vis}}, g_{\text{txt}})$ (they come from the baseline forward pass with a single joint softmax). Let

$$G_{l,h}(g_{\text{vis}}, g_{\text{txt}}) = \frac{\partial \ell_{l,h}(g_{\text{vis}}, g_{\text{txt}})}{\partial \tilde{O}_{l,h}} \in \mathbb{R}^{L_{\text{q}} \times d_h}.$$

The chain rule gives, for any $(g_{\text{vis}}, g_{\text{txt}})$,

$$\frac{\partial \ell_{l,h}}{\partial g_{\text{vis}}}(g_{\text{vis}}, g_{\text{txt}}) = \left\langle G_{l,h}(g_{\text{vis}}, g_{\text{txt}}), \frac{\partial \tilde{O}_{l,h}}{\partial g_{\text{vis}}} \right\rangle = \left\langle G_{l,h}(g_{\text{vis}}, g_{\text{txt}}), O_{l,h}^{\text{vis}} \right\rangle,$$

and analogously

$$\frac{\partial \ell_{l,h}}{\partial g_{\text{txt}}}(g_{\text{vis}}, g_{\text{txt}}) = \left\langle G_{l,h}(g_{\text{vis}}, g_{\text{txt}}), O_{l,h}^{\text{txt}} \right\rangle.$$

Evaluating at the baseline $(1, 1)$ yields

$$\frac{\partial \ell_{l,h}}{\partial g_{\text{vis}}}(1,1) = \left\langle G_{l,h}(1,1), O_{l,h}^{\text{vis}} \right\rangle, \qquad \frac{\partial \ell_{l,h}}{\partial g_{\text{txt}}}(1,1) = \left\langle G_{l,h}(1,1), O_{l,h}^{\text{txt}} \right\rangle.$$

By definition, the exact visual do-difference is

$$\Delta_{l,h}^{\text{vis}} \triangleq \ell_{l,h}(1,1) - \ell_{l,h}(0,1).$$

Consider the one-dimensional function obtained by restricting $\ell_{l,h}$ to the visual path $\tau \mapsto (\tau, 1)$, i.e., $\phi(\tau) \triangleq \ell_{l,h}(\tau, 1)$. By the fundamental theorem of calculus,

$$\Delta_{l,h}^{\text{vis}} = \phi(1) - \phi(0) = \int_0^1 \phi'(\tau)\, d\tau.$$

Since $g_{\text{txt}}$ is fixed to 1 on this path, the chain rule gives $\phi'(\tau) = \frac{\partial \ell_{l,h}}{\partial g_{\text{vis}}}(\tau, 1)$, hence

$$\Delta_{l,h}^{\text{vis}} = \int_0^1 \frac{\partial \ell_{l,h}}{\partial g_{\text{vis}}}(\tau, 1)\, d\tau.$$

This path-integral form is closely related to Integrated Gradients (Sundararajan et al., 2017): both express a finite intervention effect as an integral of a partial derivative along a continuous path. Here the path is over the gating variable, and we adopt a

single-point (right-endpoint) discretization for efficiency. We approximate this path integral by its right-endpoint value, yielding the first-order estimator

$$\widehat{\Delta}_{l,h}^{\mathrm{vis}} \triangleq \frac{\partial \ell_{l,h}}{\partial g_{\mathrm{vis}}}(1,1).$$

Using the linear gating relation $\tilde{O}_{l,h} = g_{\mathrm{vis}} O_{l,h}^{\mathrm{vis}} + g_{\mathrm{txt}} O_{l,h}^{\mathrm{txt}}$ and the definition $G_{l,h} = \partial \ell_{l,h} / \partial \tilde{O}_{l,h}$, we have

$$\frac{\partial \ell_{l,h}}{\partial g_{\mathrm{vis}}}(1,1) = \left\langle G_{l,h}(1,1), O_{l,h}^{\mathrm{vis}} \right\rangle,$$

which proves the claim. The textual case follows analogously. $\square$

*Remark* C.1 (Multi-point gate integral). Instead of the single right-endpoint approximation, one may use an $m$-point Riemann estimator $\widehat{\Delta}_{l,h}^{\mathrm{vis}}(m) = \frac{1}{m} \sum_{k=1}^{m} \frac{\partial \ell_{l,h}}{\partial g_{\mathrm{vis}}}(k/m, 1)$, which converges to $\Delta_{l,h}^{\mathrm{vis}}$ as $m$ increases. Under the same Lipschitz-type condition as Proposition B.1, the discretization error decreases on the order of $1/m$, trading additional gate-query passes for higher accuracy.

*Remark* C.2 (General paths). More generally, one can integrate along any continuous path $c(\tau) = (g_{\mathrm{vis}}(\tau), g_{\mathrm{txt}}(\tau))$ to define joint interventions, with corresponding discretizations.

# D. Implementation Details

## D.1. Benchmark Details

**MME.** MME (Fu et al., 2025) is a comprehensive benchmark for assessing general multimodal capabilities of LVLMs. It organizes evaluation into two major categories: perception and cognition. The perception category covers fine-grained visual understanding skills, including existence, count, position/location, color, poster, celebrity, scene, landmark, artwork, and OCR. The cognition category focuses on higher-level reasoning and knowledge integration, including commonsense reasoning, numerical calculation, text translation, and code reasoning. Each MME sample consists of an image-question pair, and questions are formatted in a short VQA style with binary (Yes/No) responses, enabling a broad yet unified evaluation across diverse ability dimensions.

**POPE.** Polling-based Object Probing Evaluation (POPE) (Li et al., 2023) is a binary VQA benchmark designed to assess object-level grounding and object hallucination in LVLMs. Given an image from MS-COCO and a templated query of the form "Is there a <object> in this image?", the model must answer Yes or No. We evaluate on the three standard POPE subsets—random, popular, and adversarial—each containing 3,000 questions. The three subsets differ in how negative objects are selected: (i) random samples negatives roughly uniformly from the overall vocabulary, so many negatives are less contextually plausible; (ii) popular draws negatives from frequently occurring categories, making language priors stronger and false positives more likely; and (iii) adversarial chooses negatives that are contextually plausible for the image (e.g., objects that commonly co-occur with present objects or fit the scene), making it the most challenging split. We report **Accuracy** and **F1** over Yes/No predictions (treating Yes as the positive class):

$$\mathrm{Accuracy} = \frac{\#\mathrm{correct}}{\#\mathrm{total}}, \quad \mathrm{Precision} = \frac{TP}{TP + FP}, \quad \mathrm{Recall} = \frac{TP}{TP + FN}, \quad \mathrm{F1} = \frac{2 \cdot \mathrm{Precision} \cdot \mathrm{Recall}}{\mathrm{Precision} + \mathrm{Recall}}.$$

**CHAIR.** Caption Hallucination Assessment with Image Relevance (CHAIR) (Rohrbach et al., 2018) is a benchmark for measuring object hallucination in image captioning. It prompts LVLMs to generate a description for each input image and compares the mentioned objects against the ground-truth object set (e.g., MS-COCO annotations). Using the CHAIR object lexicon to map noun phrases in the caption to object categories, an object is counted as hallucinated if it is mentioned by the model but does not appear in the ground truth. CHAIR quantifies hallucination at both the instance and sentence levels:

$$\mathrm{CHAIR}_I = \frac{|\{\text{hallucinated objects}\}|}{|\{\text{all mentioned objects}\}|}, \quad \mathrm{CHAIR}_S = \frac{|\{\text{captions with hallucinated objects}\}|}{|\{\text{all captions}\}|}.$$

Lower values indicate fewer hallucinations. To evaluate whether captions still cover the necessary visual content, we additionally report instance-level recall:

$$\mathrm{Recall} = \frac{|\{\text{non-hallucinated objects}\}|}{|\{\text{all existing objects}\}|}.$$

We also report the average response length (**Len**), computed as the mean number of generated tokens per caption, to control for potential verbosity changes across methods.

**MMHal-Bench.** MMHal-Bench (Sun et al., 2024) is a specialized benchmark for evaluating multimodal hallucination in LVLMs, consisting of 96 carefully designed image–question pairs spanning eight categories: object attributes (ATTR), adversarial objects (ADV), comparisons (COMP), counting (COUNT), spatial relations (SPAT), environmental inferences (ENV), holistic descriptions (HOL), and others (OTHER). Following prior work, model outputs are judged by GPT-4 against the reference answer and associated object information, producing an overall score $s_i$ on a 0–5 scale (higher is better). We report the **Overall Score** and **Hallucination rate (Hu.%)**:

$$\text{Score} = \frac{1}{N} \sum_{i=1}^{N} s_i, \qquad \text{Hu.\%} = \frac{1}{N} \sum_{i=1}^{N} \mathbb{I}[s_i < 3] \times 100,$$

where $N$ is the number of examples and responses with $s_i < 3$ are counted as hallucinations.

**AMBER.** An LLM-free Multi-dimensional Benchmark (AMBER) (Wang et al., 2023a) evaluates multimodal hallucinations without relying on LLM-based judges. It contains **1,004** manually curated images with structured annotations spanning object existence, attributes (e.g., state/number/action), relations (e.g., direct contact), and per-image hallucinatory target objects, covering 337 object categories. AMBER supports both generative and discriminative evaluation: in the generative setting, models produce free-form captions (e.g., "Describe this image."), and AMBER extracts mentioned objects with an official lexicon; in the discriminative setting, AMBER constructs **10,586** binary (Yes/No) probes over existence/attributes/relations and counterfactual hallucinatory targets. For generative evaluation, let $A_i$ be the ground-truth object set, $\hat{A}_i$ the extracted mentioned-object set, $H_i = \hat{A}_i \setminus A_i$ the hallucinated-object set, and $T_i$ the hallucinatory target set for image $i$. AMBER reports:

$$\text{CHAIR} = \frac{1}{N} \sum_{i=1}^{N} \frac{|H_i|}{|\hat{A}_i|}, \quad \text{Cover} = \frac{1}{N} \sum_{i=1}^{N} \frac{|\hat{A}_i \cap A_i|}{|A_i|}, \quad \text{Hal} = \frac{1}{N} \sum_{i=1}^{N} \mathbb{I}[|H_i| > 0], \quad \text{Cog} = \frac{1}{N} \sum_{i=1}^{N} \frac{|\hat{A}_i \cap T_i|}{|T_i|}.$$

For discriminative evaluation, AMBER reports standard classification scores (Acc./F1), and further summarizes overall performance with the aggregated **AMBER Score**:

$$\text{AMBER Score} = \frac{1}{2} \big( 100 - \text{CHAIR} + \text{F1} \big).$$

### D.2. Models and Experiments Setup

Most experiments are conducted on two NVIDIA A40 GPUs (48 GB memory per GPU). For the larger LLaVA-NeXT-34B and Qwen2.5-VL-32B-Instruct models, we use a single NVIDIA RTX PRO 6000 Blackwell GPU (96 GB memory).

**Model Architectures.** In Table 12, we present the detailed architectures of the LVLMs used in our experiments, where the gray rows correspond to the backbones used in our main experiments. These models are based on the Vision Transformer (ViT) (Dosovitskiy et al., 2021) and employ pre-trained vision encoders from various sources.

*Table 12.* Details of the LVLM architectures used in our experiments.

| Model | Vision encoder | LLM |
|---|---|---|
| LLaVA-1.5-7B (Liu et al., 2024b) | CLIP ViT-L/14 (336px) (Radford et al., 2021) | Vicuna-v1.5-7B (Chiang et al., 2023) |
| Qwen-VL-Chat-7B (Bai et al., 2023) | OpenCLIP ViT-bigG (Ilharco et al., 2021) | Qwen-7B |
| Qwen2.5-VL-7B-Instruct (Bai et al., 2025) | ViT w/ 2D-RoPE & window attention | Qwen2.5-7B |
| LLaVA-1.5-13B (Liu et al., 2024b) | CLIP ViT-L/14 (336px) (Radford et al., 2021) | Vicuna-v1.5-13B (Chiang et al., 2023) |
| LLaVA-NeXT-34B (Li et al., 2024) | CLIP ViT-L/14 (336px) (Radford et al., 2021) | Nous-Hermes-2-Yi-34B (Yi-34B family) |
| Qwen2.5-VL-32B-Instruct (Bai et al., 2025) | ViT w/ 2D-RoPE & window attention | Qwen2.5-32B |

**Hyperparameters.** Our hyperparameters include the intervened layer range $(L_{\text{start}}, L_{\text{end}})$, the number of selected heads $k$, and the gating strength $\gamma$. Concretely, on the POPE `popular` subset, we intervene on mid-to-upper transformer layers,

using $(L_{\text{start}}, L_{\text{end}}) = (8, 19)$ for LLaVA-1.5-7B, $(10, 24)$ for Qwen-VL-Chat, and $(9, 17)$ for Qwen2.5-VL-7B-Instruct. At each intervened layer, we select the top-$k$ heads with the smallest VRI values and apply text-route gating with strength $\gamma$. The optimal $k$ and $\gamma$ across benchmarks are summarized in Table 13.

*Table 13.* Hyperparameter search selected for $k$ and $\gamma$ across benchmarks.

| Model | parameter | POPE (random) | POPE (popular) | POPE (adversarial) | MME | CHAIR | MMHal-Bench | AMBER-G | AMBER-D |
|---|---|---|---|---|---|---|---|---|---|
| **LLaVA-1.5-7B** | $k$ | 7 | 11 | 9 | 12 | 11 | 8 | 9 | 10 |
|  | $\gamma$ | 0.6 | 0.5 | 0.5 | 0.5 | 0.7 | 0.5 | 0.6 | 0.7 |
| **Qwen-VL-Chat-7B** | $k$ | 10 | 8 | 12 | 10 | 8 | 9 | – | – |
|  | $\gamma$ | 0.4 | 0.7 | 0.5 | 0.4 | 0.6 | 0.6 | – | – |
| **Qwen2.5-VL-7B-Instruct** | $k$ | 10 | 9 | 11 | – | – | – | – | – |
|  | $\gamma$ | 0.4 | 0.6 | 0.5 | – | – | – | – | – |

### D.3. Choice of the Scalar Objective $\ell$

To compute decision-aligned route effects, we require a scalar objective $\ell$ whose gradient reflects the model's preference for the target decision at the current decoding step. We instantiate $\ell$ differently for discriminative (binary) and generative benchmarks, matching each evaluation protocol.

**Discriminative tasks (binary Yes/No).** For discriminative hallucination benchmarks such as POPE, MME and the binary subset of AMBER, the model is prompted to output *only* "Yes" or "No". Accordingly, we define the scalar target as the log-odds margin

$$\ell \;=\; \log p(\text{YES} \mid x) \;-\; \log p(\text{NO} \mid x),$$

where $p(\cdot \mid x)$ is the next-token distribution conditioned on the multimodal input $x$. This choice is both *decision-aligned* and *interpretable*: $\ell > 0$ indicates a higher preference for "Yes" than "No", while $\ell < 0$ indicates the opposite. In particular, when the candidate set is restricted to $\{\text{YES}, \text{NO}\}$, we have

$$\ell \;=\; \log \frac{p(\text{YES} \mid x)}{p(\text{NO} \mid x)},$$

so changes in $\ell$ directly correspond to how an intervention shifts the binary decision boundary. Therefore, the sign and magnitude of the estimated causal contribution under this $\ell$ admit a straightforward interpretation: a positive contribution supports the "Yes" decision, whereas a negative contribution supports "No".

**Empirical note.** We also experimented with using $\ell = \log p(y_t \mid x)$ for discriminative tasks, but found it consistently inferior to the log-odds margin $\log p(\text{YES} \mid x) - \log p(\text{NO} \mid x)$, likely because the margin explicitly accounts for the competing alternative and thus better matches the binary decision structure.

**Generative tasks (open-ended decoding).** For open-ended generation, we apply the intervention *token-by-token* and define the step-wise target as

$$\ell_t \;=\; \log p(y_t \mid x, y_{<t}),$$

where $y_t$ denotes the token selected at decoding step $t$ (under the chosen decoding rule), and $p(\cdot \mid x, y_{<t})$ is the next-token distribution. Intuitively, $\ell_t$ measures how strongly the model commits to the current token, enabling us to quantify whether the visual/textual routes *support* or *contradict* this local decision at step $t$.

Operationally, at each step $t$ we (i) run a standard one-step decode forward to obtain $p(\cdot \mid x, y_{<t})$ and the selected token $y_t$; (ii) compute the local sensitivity for this decision via a single backward pass,

$$G_t \;=\; \frac{\partial \ell_t}{\partial \tilde{O}_t},$$

where $\tilde{O}_t$ denotes the intervened route/head activations at step $t$; (iii) aggregate the resulting head-/route-level effects into $r_t$ and apply the gate; and finally (iv) re-run a lightweight one-step forward with the gate applied to produce the actual token emission. This procedure requires two forward passes and one gradient computation per generated token, but the

gradient is taken only through a *single-step* decode (with the prefix KV-cache already built), so its overhead is modest: in our profiling, the autograd component takes 8.77 ms per step, accounting for 6.30% of the total token-generation time, while computing $r_t$ and applying the gate are lightweight. Moreover, the memory footprint is close to regular decoding: we do not retain computation graphs across steps, so the backward-related activations are transient, and the dominant KV-cache growth remains essentially the same as the baseline decode.

**Token filtering and object-only intervention.**    A practical concern in token-by-token intervention is that many generated tokens are weakly semantic (e.g., punctuation such as ",", "."), for which estimating route effects may be unnecessary and could add noise. We therefore explored two variants: (i) skipping intervention on such non-informative tokens; and (ii) performing intervention only when the current token corresponds to an object word. For (ii), we leverage the CHAIR object vocabulary and apply the intervention only if the selected token $y_t$ matches an entry in this list. While this object-only strategy is straightforward on CHAIR-style evaluation where the object set is predefined, it is less practical in open-world generation because the space of possible objects is unbounded and the relevant lexicon is not known a priori. Empirically, we find that restricting intervention to object tokens yields performance close to our default full-token intervention, suggesting that the gains are largely driven by object-related decisions and that our method is not overly sensitive to intervening on punctuation or other low-content tokens.

## E. Additional Experiments

### E.1. Results on More Advanced LVLMs

To assess whether our intervention generalizes beyond the primary 7B backbones, we further evaluate it on several stronger LVLMs with different architectures and larger model scales. Table 14 reports POPE-average results on more advanced LVLMs, where each entry is *Accuracy / F1*. Overall, our method yields consistent gains across both LLaVA-style and Qwen-VL-style backbones, including larger-capacity models (LLaVA-1.5-13B, Qwen2.5-VL-32B-Instruct, and LLaVA-NeXT-34B). This suggests that the intervention does not rely on a particular model architecture or a specific pretraining recipe, but instead captures a more general property of how LVLMs mix visual and textual evidence during decision making.

Notably, the improvements persist as model scale increases. While stronger backbones typically improve general instruction-following and linguistic priors, they may still exhibit "over-confident yes" tendencies under weak visual evidence in POPE-style settings. Our method counteracts such bias by suppressing visually unsupported routes at decoding time, leading to simultaneous improvements in Accuracy and F1. Importantly, our method improves both metrics for all evaluated models, implying that it does not trade off correctness for conservativeness (or vice versa) on POPE-average. These results further support the robustness and general applicability of our approach beyond the 7B setting.

*Table 14.* **Results on more advanced LVLMs (POPE-average).** Each entry is Accuracy / F1-Score. Our method consistently improves both accuracy and F1 across larger backbones.

| Method | LLaVA-1.5-7B | Qwen2.5-VL-7B-Instruct | LLaVA-1.5-13B | Qwen2.5-VL-32B-Instruct | LLaVA-NeXT-34B |
|---|---|---|---|---|---|
| Regular | 81.37 / 79.65 | 83.99 / 84.31 | 83.05 / 82.81 | 84.25 / 85.31 | 83.17 / 84.79 |
| CRG | 87.71 / 86.61 | 89.72 / 88.55 | 87.87 / 87.94 | 89.92 / 88.93 | 89.74 / 87.73 |

### E.2. Implementation with VCD

To examine whether our conflict-aware text-route gating can be combined with existing training-free decoding heuristics, we integrate our method with VCD (Leng et al., 2024), a representative contrastive decoding strategy. Below we briefly summarize VCD and then describe a simple way to apply it on top of our method.

**Visual Contrastive Decoding.**    VCD (Leng et al., 2024) is a training-free decoding method that mitigates hallucinations by contrasting the model's token preferences under the original image and a visually degraded counterfactual. At each decoding step $t$, VCD computes logits from the original image $I$ and a perturbed image $I^-$ (e.g., blurred or noised), denoted by $z_t^+ = \text{logits}(I, y_{<t})$ and $z_t^- = \text{logits}(I^-, y_{<t})$, respectively. It then forms a contrastive logit for sampling/greedy decoding:

$$\tilde{z}_t = z_t^+ + \alpha \, (z_t^+ - z_t^-) = (1 + \alpha)z_t^+ - \alpha z_t^-,$$

where $\alpha \geq 0$ controls the contrast strength. Intuitively, tokens that remain highly probable even under the weakened visual evidence $I^-$ are more likely driven by language priors; subtracting $z_t^-$ suppresses such prior-dominant predictions and promotes image-grounded alternatives.

**VCD on top of our conflict-aware text-route gating.** We incorporate VCD on top of our Conflict A+B text-route gating by using our method as the positive branch and a regular model as the negative branch. Concretely, for the original image $I$, we enable Conflict A+B gating to obtain $z_t^{\text{CRG}} = \text{logits}(\text{CRG}(I), y_{<t})$. For the counterfactual image $I^-$, we follow the standard VCD construction but disable CRG (i.e., keep the original model unchanged) to preserve a strong language-prior reference, yielding $z_t^{\text{Reg},-} = \text{logits}(\text{Regular}(I^-), y_{<t})$. We then apply the same contrastive fusion:

$$\tilde{z}_t = z_t^{\text{CRG}} + \alpha\,(z_t^{\text{CRG}} - z_t^{\text{Reg},-}).$$

This design combines conflict-aware suppression on the original image with contrastive penalization of tokens that are insensitive to visual perturbations, providing potentially complementary signals during generation.

We evaluate the proposed combination on both LLaVA-1.5-7B and Qwen-VL-Chat over the three POPE subsets (random, popular, adversarial). As shown in Table 15, CRG$_{+\text{VCD}}$ brings only marginal changes relative to CRG: it slightly improves accuracy in most settings, while the F1 score can be comparable or slightly lower on some subsets.

*Table 15.* POPE results for LLaVA-1.5-7B and Qwen-VL-Chat, comparing REGULAR, VCD, CRG (ours) and CRG$_{+\text{VCD}}$.

| Setting | Method | LLaVA-1.5-7B | | Qwen-VL-Chat | |
|---|---|---|---|---|---|
| | | Accuracy | F1-Score | Accuracy | F1-Score |
| Random | Regular | 83.29 | 81.33 | 84.63 | 82.61 |
| | VCD | 87.73 | 87.16 | 86.93 | 85.46 |
| | **CRG (ours)** | 90.30 | **89.51** | **89.46** | 88.33 |
| | **CRG$_{+\text{VCD}}$** | **90.43** | 89.30 | 89.43 | **88.63** |
| Popular | Regular | 81.88 | 80.06 | 83.63 | 81.53 |
| | VCD | 85.38 | 85.06 | 85.17 | 83.68 |
| | **CRG (ours)** | 88.40 | **86.54** | 87.63 | **86.55** |
| | **CRG$_{+\text{VCD}}$** | **88.69** | 86.04 | **87.66** | 86.46 |
| Adversarial | Regular | 78.96 | 77.57 | 81.03 | 79.30 |
| | VCD | 80.88 | 81.33 | 83.10 | 82.04 |
| | **CRG (ours)** | 84.43 | 83.77 | 84.70 | 83.78 |
| | **CRG$_{+\text{VCD}}$** | **84.97** | **83.81** | **84.76** | **83.84** |

## E.3. Detailed Results of AMBER & MMHal-Bench

**Detailed Results of MMHal-Bench.** As reported in Table 16, our method achieves the best results on both backbones. On LLaVA-1.5-7B, we improve the score from 2.23 to 2.69 while reducing Hu.% from 65.3 to 50.9; on Qwen-VL-Chat, the score increases from 2.27 to 2.80 with Hu.% dropping from 58.2 to 48.8. Compared with the strongest baseline (VTI), our approach still yields consistent gains in both Score and Hu.%, indicating more informative generations with fewer hallucinations.

*Table 16.* Results on MMHal-Bench (evaluated by GPT-4)

| Method | LLaVA-1.5-7B | | Qwen-VL-Chat | |
|---|---|---|---|---|
| | Score↑ | Hu.%↓ | Score↑ | Hu.%↓ |
| Regular | 2.23 | 65.3 | 2.27 | 58.2 |
| VCD | 2.31 | 58.7 | 2.32 | 56.5 |
| OPERA | 2.40 | 56.4 | 2.49 | 55.3 |
| VTI | 2.51 | 53.7 | 2.62 | 51.4 |
| CRG | **2.69** | **50.9** | **2.80** | **48.8** |

*Table 18.* Supplementary POPE results on MS-COCO. Each split reports Accuracy and F1-Score.

| Method | Random | | Popular | | Adversarial | | Average | |
|---|---|---|---|---|---|---|---|---|
| | Acc. | F1 | Acc. | F1 | Acc. | F1 | Acc. | F1 |
| ONLY (Wan et al., 2025) | 89.70 | 89.10 | 86.00 | 86.31 | 79.40 | 81.07 | 85.03 | 85.49 |
| CAUSALMM (Zhou et al., 2025) | 88.93 | 88.10 | 87.13 | 87.26 | 83.70 | 82.78 | 86.59 | 86.05 |
| ICT (Chen et al., 2025) | 90.11 | 90.03 | 87.50 | 87.60 | 84.43 | 83.74 | 87.35 | 87.12 |
| DMAS (Yin et al., 2026) | – | – | – | – | – | – | 86.81 | 86.79 |
| CRG (this work) | **90.30** | 89.51 | **88.40** | 86.54 | **84.43** | **83.77** | **87.71** | 86.61 |

**Detailed Results of AMBER.**  Table 17 reports the per-metric results on AMBER for LLaVA-1.5-7B. Compared with regular decoding and PAI, our method substantially reduces object hallucination (CHAIR 8.3 to 4.6, Hal 36.7 to 23.2) while improving coverage and decision quality (Cover 49.4 to 53.3, F1 73.7 to 77.5). As a result, it achieves the best overall AMBER score (82.70 to 86.45).

*Table 17.* Results on AMBER (LLaVA-1.5-7B). The AMBER metric is computed as $(100 - \text{CHAIR} + F1)/2$.

| Model | Method | CHAIR↓ | Cover↑ | Hal↓ | Cog↓ | Acc.↑ | Prec.↑ | Rec.↑ | F1↑ | AMBER↑ |
|---|---|---|---|---|---|---|---|---|---|---|
| | Regular | 8.3 | 49.4 | 36.7 | 4.4 | 72.7 | 84.7 | 62.1 | 73.7 | 82.70 |
| LLaVA-1.5-7B | PAI | 6.7 | 48.8 | 42.3 | **1.9** | 74.3 | 87.2 | **73.8** | 74.8 | 84.05 |
| | CRG | **4.6** | **53.3** | **23.2** | 2.1 | **77.4** | **92.3** | 72.1 | **77.5** | **86.45** |

### E.4. Supplementary POPE Comparisons with Recent Methods

Table 18 also compares CRG with recent training-free hallucination mitigation methods on the standard POPE benchmark under the MS-COCO setup. These methods include ONLY (Wan et al., 2025), CAUSALMM (Zhou et al., 2025), ICT (Chen et al., 2025), and DMAS (Yin et al., 2026). The Average columns are arithmetic means over the three standard POPE splits: Random, Popular, and Adversarial. The CRG row uses the results reported in this submission.

**Relation to process- and attention-based signals.**  Recent work also suggests that process-level or attention-based signals can be useful. For example, Knowledge Transfer from Interaction Learning (Gao et al., 2025) emphasizes the value of interaction processes rather than only final representations. In hallucination mitigation, Modality Bias in LVLMs (Zheng & Zhang, 2025) uses attention-lens analysis to rebalance modality usage, and AdaIAT (Zhong et al., 2026) adaptively modifies attention to generated text to reduce hallucination while preserving fluency. These directions are complementary to CRG. Our claim is not that attention or interaction signals are useless; rather, attention allocation alone is insufficient as the primary decision-aligned criterion for selective gating. CRG uses route effects to determine whether a route helps or hurts the current decision, while attention/process signals can provide complementary diagnostics or future triggers.

## F. Limitations and More Discussion

### F.1. Scope Boundary of CRG

CRG should be read as a conflict detector and intervention, not as a general hallucination corrector. It targets cases in which visual evidence is available, but a prior-driven text route dominates the current token decision. This is the setting our route decomposition can identify and act on: the visual and text routes push the decision in opposite directions, and suppressing the selected text route can restore a more visually grounded choice. Errors whose cause does not pass through such visual-text route conflict are outside the primary mechanism.

The key non-target case is same-direction bias, which is different from the target failure mode addressed by CRG. CRG is designed for cases where usable visual evidence is present in the model but is overridden at the token decision by a stronger prior-driven text route. In this regime, the visual and text routes push in opposite directions, giving CRG a decision-aligned conflict signal: suppressing the selected text route allows the existing visual evidence to carry more influence. By contrast, if both routes already support the same wrong answer, CRG has no internal conflict signal to exploit. For example, in a rare-attribute case such as a pink banana, the visual representation may itself collapse to a stereotyped "yellow banana"

feature, while the language prior also favors "yellow banana". CRG cannot create missing visual counter-evidence or flip a visual route that is itself biased; such cases are missed-correction cases caused by deficient or biased visual encoding, rather than failures of the intended conflict-aware gating mechanism. This distinction is why we do not make a universal claim over OCR, reasoning, or all attribute errors: CRG is expected to help when the error is mediated by a text-over-vision route conflict.

A second boundary appears when visual evidence is weak or ambiguous. In these cases, language priors may provide useful contextual guesses, whereas CRG tends to suppress prior-driven affirmative answers when estimated visual support is weak. The degraded-evidence case study in Appendix H illustrates this behavior. We therefore view this regime as a faithfulness–helpfulness tradeoff: CRG favors calibrated uncertainty and grounded faithfulness over unsupported, but sometimes useful, guessing.

This scope also determines how to interpret results on broader hallucination categories. MME, MMHal-Bench, and AMBER include attributes, colors, spatial relations, OCR-sensitive categories, and higher-level multimodal judgments; CRG is expected to help on these categories only when the error is mediated by visual-text route conflict. Errors dominated by poor perception, OCR, weak visual representations, or multi-step reasoning require complementary upstream or reasoning-oriented methods. Finally, CRG is a white-box inference-time intervention requiring access to activations, gradients, and route-level patching, so it is aimed at open-weight or self-hosted LVLMs rather than closed-source API-only systems. Adapting the idea to black-box settings would require output-level proxies, such as sensitivity under image/text perturbations, and is left for future work.

### F.2. Cross-Gate Interactions

CRG ranks heads using local first-order route effects, but the final intervention applies multiple gates jointly. This creates the possibility of higher-order interactions: suppressing one text route can change downstream hidden states, which may in turn change the effect of another gate in a later layer. Thus, the first-order theory justifies the local ranking and selection signals, but it does not fully characterize all cross-gate couplings during the full autoregressive computation.

A local view makes this limitation explicit. Let $g$ collect the selected gate values and let $\ell(g)$ denote the decision score after applying these gates. Around the unmodified model $g = \mathbf{1}$, we can write

$$\ell(g) - \ell(\mathbf{1}) \approx \sum_i \left.\frac{\partial \ell}{\partial g_i}\right|_{\mathbf{1}} (g_i - 1) + \frac{1}{2} \sum_{i \neq j} \frac{\partial^2 \ell}{\partial g_i \partial g_j}(\tilde{g})(g_i - 1)(g_j - 1) + \cdots.$$

The first-order terms are the local signals used by CRG for ranking and selection. The cross terms correspond to the concern that one gate may change the effect of another, especially across layers after the hidden state has already been modified.

We control this interaction risk through design choices that reduce both the number and the magnitude of possible cross terms, although they do not make the interaction terms vanish. First, the intervention is sparse: CRG selects only a small number of low-VRI heads within the chosen layer range, instead of gating all conflicting heads. Since the second-order sum involves pairs of selected gates, sparsity directly limits how many gate–gate interaction terms can contribute. This also makes the intervention easier to monitor empirically, because failures caused by interactions would have to arise from a small selected set rather than from dense global suppression.

Second, each selected gate is route-specific and graded. CRG modifies only the text-route contribution of a selected head, leaving the visual route and the rest of the head computation intact. Thus, the perturbation is not a full head deletion but a partial change of the form $(g_i - 1)O_i^{\text{txt}}$. In the cross term, this makes the effective interaction scale depend on the product of two graded deviations, $(g_i - 1)(g_j - 1)$, and on the text-route components being attenuated. This is substantially milder than removing entire heads, which would perturb both visual and text routes and could introduce larger downstream state shifts.

Third, the effective gates concentrate in a contiguous mid-to-upper layer block. This matters because perturbations in very early layers can propagate through many later transformations before another gate is applied, creating more opportunities for long-range accumulation. By contrast, a localized mid-to-upper block applies the intervention after lower-level visual features have already been formed and over a shorter downstream path. The gates can still interact within this block, but the design avoids broad, all-layer perturbations and reduces the chance that early route changes cascade through the full decoder before later gates act.

Empirically, strong negative complementarity would likely appear as brittle sensitivity to $k$, $\gamma$, or layer range, or as the

full Conflict A+B intervention underperforming a single-branch variant. Instead, Table 5 shows that the combined A+B intervention is consistently strongest, while the layer-range and hyperparameter sweeps in Figures 6 and 7 show smooth behavior. We therefore view cross-gate interaction as a real but secondary limitation: it is not fully eliminated by the current formulation, but the sparse and graded design keeps it controlled in practice.

### F.3. Per-Token CRG in Open-Ended Generation

For open-ended generation, CRG is not based on a single attribution computed once and then reused for the entire answer. Instead, it is recomputed at each decoding step $t$: we define

$$\ell_t = \log p(y_t \mid x, y_{<t}),$$

estimate visual/text route effects for the current token, apply conflict-aware gating, and then proceed to the next step. The final answer is therefore shaped by a sequence of local decision-time interventions.

At the same time, not every generated token is equally tied to the final grounded semantic decision. In a sentence such as "It is a lovely dog," content-bearing tokens such as "dog" are more directly grounded in the image, whereas function words such as "It", "is", and "a" mainly support linguistic realization and fluency. This raises a valid concern: applying CRG at every token may introduce intervention on positions that are only weakly related to the final grounded content.

We examined three practical strategies on CHAIR using LLaVA-1.5-7B with `max_new_tokens`=128. The first is an oracle-like vocabulary-triggered strategy that applies CRG only when the current token matches the closed CHAIR object vocabulary, i.e., the MS-COCO object categories and their synonyms. The second is our default step-wise CRG, which applies CRG at every decoding step while still selecting heads and gates based on the current token's route conflict. The third is a first-token-only variant motivated by the observation that early token distributions can contain response-level signals.

*Table 19.* Comparison of token-level intervention strategies on CHAIR with LLaVA-1.5-7B. Lower $C_S$ and $C_I$ are better; higher recall is better.

| Setting | $C_S \downarrow$ | $C_I \downarrow$ | Recall↑ | Generalizable? |
|---|---|---|---|---|
| CHAIR-vocabulary-triggered CRG | 33.4 | 10.6 | 78.6 | No |
| Step-wise CRG (ours) | 34.2 | 11.2 | 77.8 | Yes |
| First-token-only CRG | 45.3 | 14.2 | 77.1 | Yes |

The vocabulary-triggered variant is slightly stronger than step-wise CRG, which is expected because it uses benchmark-specific knowledge about which tokens are evaluated by CHAIR. However, the gap is small, suggesting that the most important hallucination-related conflicts are concentrated near content-bearing positions even when CRG is evaluated at every step. The first-token-only variant is more efficient and still meaningful, but it underperforms the two token-adaptive strategies, indicating that later decoding positions remain important because hallucinated objects and attributes often emerge after the first token. We therefore use step-wise CRG as the default because it is general across open-ended generation settings without relying on benchmark-specific vocabularies, while acknowledging that more selective semantic triggering is a promising future direction.

## G. Computational Cost Analysis

Table 20 reports a *per-decoding-step* runtime breakdown on a CHAIR dialogue with `max_new_tokens`=512 using LLaVA-1.5-7B on a single A40 GPU under PyTorch eager attention. Each entry reports the average per-decoding-step time of each module and its percentage share (averaged over the generated steps).

The per-step cost is dominated by standard model execution. The baseline forward pass (`base_forward`, 39.58%) and the gated forward pass (`gated_forward`, 42.83%) together account for 82.41% of the time (114.73 ms/step), reflecting that CRG performs an additional forward pass during decoding. Estimator-specific overhead is modest: computing local sensitivities (`grad`, 6.30%) and aggregating head-route effects (`compute_CRE`, 2.84%) sum to 9.14% (12.72 ms/step). The gate computation itself is lightweight (`gate_compute`, 1.12%; 1.56 ms/step), and the remaining overhead is grouped into `other` (7.34%).

Overall, the breakdown confirms that the runtime overhead mainly comes from standard forward computation shared across methods, while the additional gradient- and CRE-related components remain a small fraction of the per-step latency.

*Table 20.* Per-decoding-step runtime breakdown on a single CHAIR dialogue using **LLaVA-1.5-7B** on a single A40 GPU with `max_new_tokens`=512 under PyTorch eager attention. Each entry reports the average time per decoding step (averaged over the generated steps) and its percentage share. The total matches the per-step latency of CRG reported in Table 6.

| Block | Avg (ms/step) | Share (%) |
|---|---|---|
| base_forward | 55.10 | 39.58 |
| grad | 8.77 | 6.30 |
| compute_CRE | 3.95 | 2.84 |
| gate_compute | 1.56 | 1.12 |
| gated_forward | 59.63 | 42.83 |
| other | 10.22 | 7.34 |
| **Total** | **139.23** | **100.00** |

**Runtime and memory overhead.** Table 21 compares average latency (reported *per decoding step*) and peak GPU memory under the same decoding setup for LLaVA-1.5-7B. CRG achieves the best overall performance, improving POPE accuracy from 81.37 to 87.71 and reducing CHAIR $C_S$ from 52.8 to 34.2. In terms of efficiency, CRG incurs a $2.27\times$ latency increase (139.2 ms/step), which is close to VCD ($2.01\times$; 123.2 ms/step) and M3ID ($2.03\times$; 124.5 ms/step), while OPERA and HALC are substantially slower ($7.12\times$ and $6.52\times$, respectively). Notably, CRG introduces essentially no additional peak memory overhead ($\times 1.00$; 14950 MB), whereas OPERA and HALC increase memory usage by more than $1.5\times$.

*Table 21.* Runtime and peak GPU memory overhead under the same decoding setup for LLaVA-1.5-7B.

| Method | Avg. Latency↓ | GPU Memory↓ | POPE-Average↑ | CHAIR $C_S$↓ |
|---|---|---|---|---|
| Regular | 61.3 ms ($\times 1.00$) | 14945 MB ($\times 1.00$) | 81.37 | 52.8 |
| VCD (Leng et al., 2024) | **123.2 ms** ($\times 2.01$) | 15749 MB ($\times 1.05$) | 84.66 | 51.6 |
| M3ID (Favero et al., 2024) | 124.5 ms ($\times 2.03$) | 15575 MB ($\times 1.04$) | 85.39 | 48.3 |
| OPERA (Huang et al., 2024) | 435.7 ms ($\times 7.12$) | 22706 MB ($\times 1.52$) | 85.69 | 44.6 |
| HALC (Chen et al., 2024) | 399.4 ms ($\times 6.52$) | 23084 MB ($\times 1.54$) | 86.32 | 39.7 |
| **CRG** | 139.2 ms ($\times 2.27$) | **14950 MB** ($\times 1.00$) | **87.71** | **34.2** |

# H. Case Studies

Figure 8 serves as an illustrative case study that isolates the causal role of visual evidence by constructing controlled counterfactual/ambiguous variants: panel (a) is an author-taken photo, while panels (b–d) are generated with Gemini 3 Pro Image (Google DeepMind, 2025) by removing the queried object, replacing it with a visually similar distractor, or degrading the evidence via blur. These examples are not included in quantitative evaluation. In our framework, the decision is summarized by the logit gap $m = \ell_{\text{yes}} - \ell_{\text{no}}$, and we estimate per-head, per-route causal gains $(\hat{\Delta}_{l,h}^{\text{vis}}, \hat{\Delta}_{l,h}^{\text{txt}})$ as interventional changes in $m$ under route-specific do-operations. Intuitively, when the image truly contains sheep (a), both routes tend to support the same decision so the estimated gains are sign-aligned and we keep the head unchanged. In contrast, in the counterfactual no-sheep setting (b) and the horse-as-distractor setting (c), regular decoding still produces an affirmative answer, which is consistent with a modality conflict pattern where the text route pushes $m$ upward ($\hat{\Delta}_{l,h}^{\text{txt}} > 0$) despite weak or negative visual support ($\hat{\Delta}_{l,h}^{\text{vis}} \leq 0$). Our intervention resolves this conflict by attenuating only the text route (via $g_{l,h}^{\text{txt}} \in [0, 1]$), thereby reducing unsupported increases in $m$ and correcting false positives. Finally, under ambiguous evidence (d), we observe the same mechanism: regular decoding commits to "yes", while our method becomes conservative because the estimated visual support is weak and conflicting, so suppressing the conflicting text-route prevents an over-confident affirmative claim.

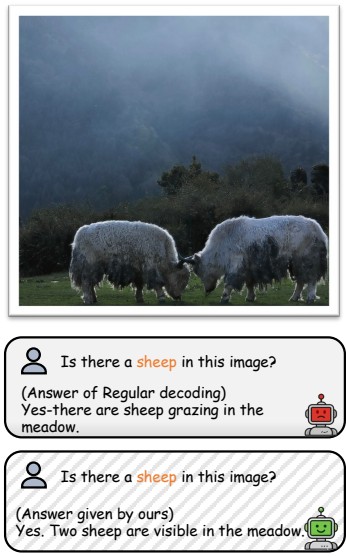

*(a)* True positive preserved: sheep present; both methods answer correctly.

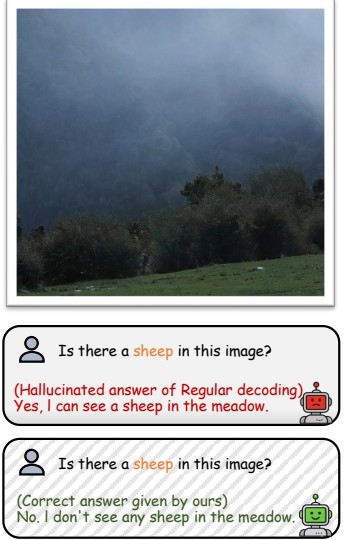

*(b)* False positive corrected: no sheep in the image; regular decoding hallucinates "yes".

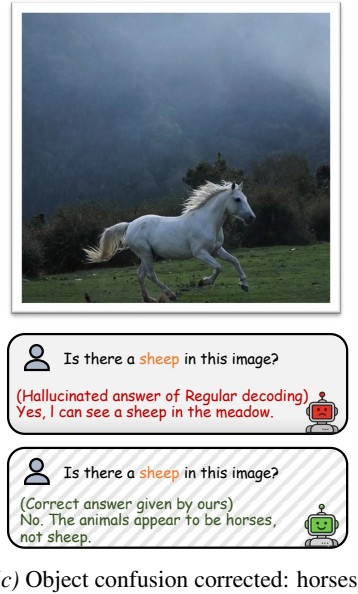

*(c)* Object confusion corrected: horses are mistaken as sheep by regular decoding.

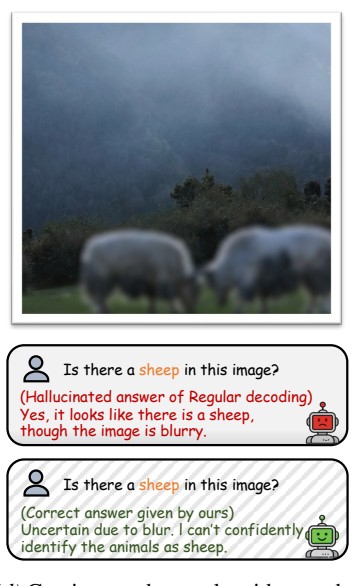

*(d)* Cautious under weak evidence: the image is blurry; regular decoding still answers "yes", while our method expresses uncertainty instead of making an unsupported claim.

*Figure 8.* Case study with real and AI-generated images. Panel (a) is an author-taken photograph, while panels (b–d) are generated with Gemini 3 Pro Image (Google DeepMind, 2025) to create controlled counterfactual or ambiguous variants for visualization. Regular decoding shows a tendency toward unsupported affirmative answers and object confusion, whereas our method suppresses over-confident "yes" responses and is conservative under weak evidence.

