# OpenReview forum: "Mitigating Hallucinations in Large Vision-Language Models via Causal Route Gating"
_ICML.cc/2026/Conference — ICML 2026 spotlight_

### Official Review · Reviewer_zkFn · 2026-02-22

**Soundness:** 3
**Presentation:** 3
**Significance:** 3
**Originality:** 3
**Overall Recommendation:** 5
**Confidence:** 4

**Summary:**

This study addresses the problem of generative illusions caused by language prior dominance in large visual language models (LVLMs) by proposing a training-free inference-time intervention framework—Causal Route Gating (CRG). This method decouples visual and textual information flows within the attention head, uses causal effect estimation to identify "illusion heads" that are dominated by language priors and conflict with visual evidence, and selectively suppresses their textual paths. This forces the model to regress to visual evidence while preserving the integrity of the visual path, achieving significant suppression of illusion generation across models and tasks with almost no increase in GPU memory overhead.

**Compliance With Llm Reviewing Policy:**

Affirmed.

**Final Justification:**

I love this paper, the best paper in my batch. But I think more discussion should be made in the final version about the newly added references, therefore, I cannot give you 6 points.

**Key Questions For Authors:**

See the Strengths And Weaknesses.

**Limitations:**

yes

**Strengths And Weaknesses:**

Overall, I believe this work presents an interesting and effective approach to mitigating hallucination in multimodal large language models. However, I have several concerns that need to be addressed before the paper can be considered for acceptance.

First, regarding the issue of inference cost (I think most reviewers would have this concern), although the authors achieved some performance gains, the inference cost seems to have increased? I think it might be possible to add some comparisons, such as trying your approach on a diffusion-based language model to avoid this problem.

Secondly, there's the issue of black boxes. Of course, I think this doesn't have to be resolved; it's just a concern.

Finally, I don't believe that VAR solutions are completely ineffective, because according to some interpretable work, VLM's concept acceptance itself is based on interaction, such as this work: "Gao Y, Chen K, Peng Z, et al. Knowledge Transfer from Interaction Learning[C]//Proceedings of the IEEE/CVF International Conference on Computer Vision. 2025: 3585-3595." Therefore, I believe a better decision-making approach should be comprehensive, considering not only cognitive outcomes (route) but also cognitive processes (attention).

---

> ### Author Rebuttal · Authors · 2026-03-30
>
> ### W1. Cost Issue and suggestion on Diffusion-based Language Models
> We thank the reviewer for raising the cost issue. We agree that CRG increases inference latency relative to regular decoding; this is the cost of route-effect estimation and conflict-aware intervention at inference time. However, this overhead is already carefully quantified in the paper **(Table 6 & 19)** and remains moderate relative to prior training-free baselines. Under the same setup, CRG has 2.27× latency with essentially no additional peak GPU memory overhead (14950 MB, 1.00×), compared with 2.01× / 1.05× for VCD, 7.12× / 1.52× for OPERA, and 6.52× / 1.54× for HALC. **Table 18** further analyzes where these computational costs come from and how they are controlled.
>
> The paper also analyzes where this overhead comes from: it is dominated by the additional forward pass rather than by an expensive search or optimization routine. For a more detailed breakdown, please also see our response to **Reviewer XBUd / cDXQ** on computational overhead, where we report the per-step runtime decomposition.
>
> Regarding the reviewer’s suggestion on diffusion-based language models, we thank the reviewer for this interesting suggestion. We agree that a diffusion-based formulation is a meaningful future direction. In principle, one could replace the current autoregressive objective $\ell_t=\log p(y_t\mid x,y_{<t})$ with a denoising-step objective, e.g.
> $$
> \ell_\tau = \sum_{i\in S_\tau}\log p_\theta(\hat y_i^\tau \mid z_\tau, I, \tau),
> $$
> and then estimate route effects and apply conflict-aware gating at selected denoising steps in the same spirit as CRG. This could potentially amortize intervention over token blocks rather than over every autoregressive token.
>
> At the same time, this would require a nontrivial reformulation rather than a drop-in replacement. Early denoising states are less semantically stable, so route-effect estimation and gate scheduling would need to be redesigned. Moreover, diffusion does not automatically remove the compute issue: repeated denoising steps may still make intervention expensive, even if the cost profile differs from autoregressive decoding. We therefore view this as a promising extension of CRG and an interesting direction for future work.
>
> ### W2. Black-Box Settings
> We agree that this is a reasonable concern. CRG is a white-box inference-time intervention, so it is not directly applicable to closed-source API-only settings. We view this mainly as a deployment-scope limitation rather than a flaw in the proposed mechanism, since our paper is explicitly situated in the open-weight/self-hosted LVLM setting. Extending the idea to black-box settings via output-level proxies would be an interesting future direction.
>
> ### W3. Attention-Based Signals
>
> We thank the reviewer for this thoughtful comment. We agree that attention-/interaction-based signals can be informative, and this is not in conflict with our paper. **Appendix A** already gives a more nuanced picture: on POPE-popular, **Table 8** shows that VAR has a moderate correlation with decision-aligned quantities ($\rho=0.5766$ with $|\hat{\Delta}^{vis}|$ and $\rho=0.5300$ with $\widehat{\mathrm{VRI}}_{l,h}$), while **Proposition A.4** formalizes a sufficient condition under which VAR can indeed be informative.
>
> Our claim is narrower: attention-mass proxies such as VAR are not sufficient as the primary decision-aligned criterion for selective hallucination intervention. Attention allocation alone cannot, in general, determine whether the attended visual content supports or opposes the current decision. Consistently, **Theorem A.6** shows that attention-only proxies cannot identify the sign of decision-aligned visual influence in general, and **Table 9** shows that even among the top-10 VAR heads, negative visual effects remain common ($\Pr(\hat{\Delta}^{vis}<0 \mid VAR \in \text{top 10 per-cent})=0.515$).
>
> We therefore agree that combining process-level signals with outcome-level signals is a promising direction. The reviewer-cited interaction-learning work [1] supports the value of process signals. More directly related hallucination-mitigation works also suggest the same: Modality Bias in LVLMs [2] already moves beyond plain VAR by jointly using TAR and VAR to rebalance textual/visual attention, and AdaIAT [3] shows that attention-based intervention can help when made adaptive. We view these directions as complementary to CRG: our emphasis on CRE/VRI is that, for selective gating, we need to know not only where attention is allocated, but whether a route is actually helping or hurting the current decision.
>
> **Reference**
>
> [1] Knowledge Transfer from Interaction Learning. ICCV 2025.
>
> [2] Modality Bias in LVLMs: Analyzing and Mitigating Object Hallucination via Attention Lens. arXiv 2025.
>
> [3] AdaIAT: Adaptively Increasing Attention to Generated Text to Alleviate Hallucinations in LVLM. arXiv 2026.

---

> > ### Author Rebuttal · Reviewer_zkFn · 2026-04-02
> >
> > I love this paper, the best paper in my batch. But I think more discussion should be made in the final version about the newly added references, therefore, I cannot give you 6 points. However, I still wish you success in your research.

---

> > > ### Author Response · Authors · 2026-04-02
> > >
> > > Thank you very much for your thoughtful follow-up and for updating your assessment. We are truly glad that our rebuttal addressed your concerns. We also sincerely appreciate your suggestion regarding the newly added references, and we will make sure to discuss them more clearly in the final version.

---

### Official Review · Reviewer_cDXQ · 2026-03-11

**Soundness:** 2
**Presentation:** 4
**Significance:** 3
**Originality:** 2
**Overall Recommendation:** 3
**Confidence:** 4

**Summary:**

This article seeks to present a central context for understanding hallucinations in large vision–language models (LVLMs), arguing that hallucination often arises from route competition between visual evidence and language priors. A significant issue considered by the article is that, even when visual tokens receive attention, the final token prediction can still be dominated by the textual pathway, leading the model to generate plausible but visually unsupported content.

To address this problem, the authors propose Causal Route Gating (CRG), a training-free inference-time intervention that decomposes each attention head into a visual route and a text route. The method estimates token-level causal effects of these routes using a lightweight one-forward-one-gradient approximation, identifies heads where visual and textual contributions conflict, and selectively suppresses the text route in heads dominated by language priors while preserving the visual route.

Experiments on several hallucination benchmarks (e.g., POPE, CHAIR, MMHal-Bench, AMBER) and multiple LVLMs (e.g., LLaVA-1.5 and Qwen-VL) show that CRG consistently reduces hallucination-related errors while maintaining overall multimodal performance with modest inference-time overhead.

**Compliance With Llm Reviewing Policy:**

Affirmed.

**Final Justification:**

I thank the authors for their effort and clarifications. However, my core concerns remain unresolved. I will maintain my score.

**Key Questions For Authors:**

1) Reliability of visual signals.
The proposed method assumes that hallucinations mainly arise from excessive reliance on language priors and therefore suppresses the text route in conflicting heads. However, visual representations themselves may encode strong dataset biases (e.g., bananas are typically yellow), and visual features may not fully disentangle object–attribute correlations. How does the method behave in cases where the visual modality itself carries biased or ambiguous signals, such as unusual object attributes (e.g., a pink banana) that are rare in the training distribution?

2) Balancing visual evidence and language priors.
Language priors can sometimes provide useful contextual information when visual evidence is weak or partially occluded. How does the proposed gating mechanism balance faithfulness to visual evidence with helpful linguistic priors in such scenarios? Have the authors observed cases where suppressing the text route leads to degraded generation quality or overly conservative outputs?

3) Sensitivity to hyperparameters and head selection.
The method involves several design choices, such as the number of heads selected for intervention (top-k), the gate ranges for different conflict types, and the intervention layer range. How sensitive is the performance to these hyperparameters across different models and tasks?

4) Generalization to other hallucination types.
The experiments mainly focus on object-level hallucinations or grounding-related errors. Could the authors comment on whether the proposed framework could also help mitigate other hallucination types, such as attribute hallucination, OCR errors, or reasoning-based hallucinations?

5) Applicability in black-box settings.
The method requires access to internal activations and gradients during inference. Do the authors foresee ways to approximate or adapt the proposed intervention in black-box or API-based settings, where such internal access is not available?

**Limitations:**

yes

**Strengths And Weaknesses:**

### Strengths

1) Clear mechanistic insight. The paper provides an intuitive explanation for hallucinations in LVLMs by attributing them to competition between visual evidence and language priors within attention heads. This route-level perspective offers a more interpretable view of how hallucinations arise during token generation.

2) Fine-grained intervention strategy. The proposed Causal Route Gating (CRG) selectively suppresses only the text route of a subset of conflicting heads while preserving the visual route. Compared with coarse head pruning or global rescaling strategies, this intervention is more targeted and less likely to disrupt normal multimodal reasoning.

3) Training-free and broadly applicable. The method operates entirely at inference time without additional training, making it relatively easy to apply to existing LVLMs.

### Weakness

1) Strong reliance on visual signals. The method assumes that hallucinations mainly arise from excessive language priors and therefore suppresses the text route in conflicting heads. However, this implicitly assumes that the visual signal is always more reliable, which may not hold in practice. In many cases, visual representations themselves can encode strong dataset priors (e.g., bananas are typically yellow), and visual features may not fully disentangle object–attribute correlations. As a result, the model may still produce biased predictions when unusual visual attributes (e.g., a black banana) are present.

2) Limited handling of visual ambiguity. When visual evidence is weak, occluded, or inherently ambiguous, suppressing the text route may lead to overly conservative or degraded predictions, since language priors can sometimes provide useful contextual information.

3) White-box assumption. The method requires access to internal activations and gradients during inference. This limits its applicability in practical settings where LVLMs are accessed through closed-source APIs.

4) Inference overhead. The approach requires additional gradient computation and head-level intervention during decoding, which increases inference latency compared with standard decoding.

5) Scope of hallucination types. The method primarily targets visual grounding errors caused by language priors, and it is unclear how well it generalizes to other hallucination types such as reasoning errors, OCR mistakes, or instruction-following failures.

---

> ### Author Rebuttal · Authors · 2026-03-30
>
> ### W1 & Q1. Visual Bias
> We thank the reviewer for this important question. We agree that visual representations can themselves be biased or ambiguous, especially for rare attributes. But CRG does not globally “trust vision”: it intervenes only on conflict heads where visual and text routes disagree, and suppresses only the most text-dominant ones (lowest VRI), leaving agreement heads unchanged. The reviewer’s rare-attribute example helps clarify the scope:
>
> pink banana -> visual representation -> yellow banana
>
> pink banana -> language prior -> yellow banana (because bananas are typically yellow)
>
> Here both routes are biased toward yellow, so this is not a Conflict-A/B case and would not trigger CRG. We agree that CRG may fail to correct this such same-direction bias from imperfect visual perception, but this is better viewed as a missed-correction case outside the primary target scope, rather than evidence that CRG assumes vision is always correct or systematically worsens such errors.
>
> More broadly, related work has begun to mitigate such errors from upstream directions, for example by defending against hallucination in the visual encoder [1]. We view CRG as complementary to these efforts: improving visual representations upstream can reduce one source of error, while CRG mitigates residual language-prior override downstream through conflict-aware route intervention at decoding time.
>
> [1] SHIELD: Suppressing Hallucinations In LVLM Encoders via Bias and Vulnerability Defense. ICLR 2026
>
> ### W2 & Q2. Visual Ambiguity.
> We agree that this is a real limitation: under weak or occluded visual evidence, CRG can make the model more conservative, as noted in our limitations and illustrated in **Fig. 8(d)**. This is a deliberate tradeoff: when grounding is weak, we prefer calibrated uncertainty over a confident but unsupported guess. CRG acts only on conflict heads and leaves agreement heads unchanged. Please also see our response to **Reviewer m64C (Q1)** for a more detailed discussion of this faithfulness-versus-helpfulness tradeoff.
>
> ### W3 & Q5. White-box assumption.
> We agree that CRG is a white-box inference-time intervention (as stated in our limitation) and thus not directly applicable to closed-source API-only settings. We view this as a deployment-scope limitation rather than a technical flaw, since our paper studies the open-weight regime common to many mechanistic and intervention-based methods.
>
> Thus, this concern affects applicability rather than validity in our target setting; extending CRG to black-box APIs via output-level sensitivity proxies under image/text perturbations is a possible future direction, but beyond the current formulation.
>
> ### W4. Inference overhead.
> Regarding computational overhead, we agree that this is an important practical concern. CRG does introduce extra latency, but **Table 6 & 19** shows that this overhead remains moderate: 2.27× latency with essentially no additional peak GPU memory overhead (1.00×), compared with 7.12× / 1.52× for OPERA and 6.52× / 1.54× for HALC. **Table 18** further analyzes where these computational costs come from and how they are controlled. Please also see our response to **Reviewer XBUd (W1)** for a more detailed runtime breakdown.
>
> ### Q3. Hyperparameter
>
> Our results do not indicate strong hyperparameter fragility. **Fig. 7** shows smooth sensitivity to $k$ and $γ$: moderate values perform best, and larger $k$ mainly brings diminishing returns. **Fig. 6** shows that effective intervention layers form a stable mid-to-upper block across models. **Table 13** further indicates that the selected $k/γ$ values vary across models/tasks, but remain within a relatively narrow, interpretable range. The conflict-specific gate ranges are semantically motivated, and **Table 5** shows that removing either branch is consistently worse than the full A+B method. While more adaptive schedules are worth exploring, the current evidence suggests robustness rather than fragility.
>
> ### W5 & Q4. Hallucination Types.
> We respectfully believe the empirical scope is broader than characterized in the review. Beyond POPE/CHAIR, we also evaluate on MME, MMHal-Bench, and AMBER, which cover attributes/colors, spatial relations, and higher-level multimodal categories. In particular, MME includes Color/OCR as well as Commonsense Reasoning and Numerical Calculation; MMHal-Bench includes ATTR and SPAT; and AMBER explicitly covers attributes and relations.
>
> Mechanistically, CRG is also not object-specific: it acts on visual/text route conflict and suppresses prior-dominant text routes when they override image-grounded evidence. Therefore, it can naturally extend beyond object existence to other grounded decisions such as attributes and relations when the same failure mode arises. We are more cautious about OCR- and reasoning-related errors: CRG may help only in the subset of such cases that are still mediated by visual-text conflict, rather than by pure perception or reasoning failure.

---

> > ### Author Rebuttal · Reviewer_cDXQ · 2026-04-03
> >
> > The rebuttal clarifies that the proposed method does not assume visual signals are always reliable, but instead focuses specifically on resolving conflicts between visual and textual routes. This addresses part of the concern regarding over-reliance on the visual modality and helps better define the intended scope of the approach. The additional discussion on efficiency and empirical coverage is also helpful.
> >
> > However, the method fundamentally operates under a restricted setting where hallucination arises from disagreement between modalities. It does not address cases where both visual and textual representations share similar biases (e.g., dataset-level correlations), which remain an important source of hallucination. As a result, the range of failure modes that can be mitigated appears limited.
> >
> > In addition, the reliance on internal model access (activations and gradients) and the focus on a specific class of hallucination errors raise concerns about general applicability across broader multimodal scenarios and practical deployment settings.
> >
> > Overall, the work is well-motivated and provides a clear mechanistic perspective, but the limitations in handling visual bias and the constrained scope reduce its overall impact. I therefore slightly increase my score but maintain a weak reject recommendation.

---

> > > ### Author Response · Authors · 2026-04-04
> > >
> > > Thank you again for the thoughtful follow-up and for clarifying the remaining concern. We agree that CRG does not directly address every hallucination source. In particular, same-direction bias, where both visual and textual routes support the same incorrect prediction, is an important but distinct failure mode outside the primary scope of the current mechanism.
> > >
> > > To clarify the remaining point of disagreement, we view the limitation here mainly as one of scope and deployment breadth, rather than the validity of the mechanism in its target setting. CRG is designed for conflict-mediated hallucination in open-weight LVLMs, and within that scope it provides a mechanistically grounded, training-free intervention with consistent gains across multiple models and benchmarks, including settings beyond pure object existence (e.g., attributes and spatial categories). Similarly, the white-box requirement limits direct applicability to API-only systems, but we view this as a deployment constraint that is common in this line of mechanistic intervention work.
> > >
> > > We will make this scope boundary more explicit in the final version, and present CRG more clearly as a targeted and complementary intervention rather than a universal solution to all LVLM hallucination types.

---

### Official Review · Reviewer_XBUd · 2026-03-13

**Soundness:** 3
**Presentation:** 3
**Significance:** 3
**Originality:** 3
**Overall Recommendation:** 5
**Confidence:** 3

**Summary:**

The paper proposes a causal route gating method to identify conflicts between visual and text routes and performs targeted interventions to alleviate hallucinations in LVLMs.

**Compliance With Llm Reviewing Policy:**

Affirmed.

**Final Justification:**

The rebuttal addresses my concerns, and I raise the score to 5.

**Key Questions For Authors:**

1. The paper alleviates hallucinations arising when language priors override visual inputs. Will this assumption miss some other types of hallucinations, such as those caused by imperfect visual perception?
2. The heads are identified from a single current token prediction. Does this reliably reflect the reasoning for the final answer the model will produce?
3. How does the method compare to recent training-free methods such as [1]?

**Limitations:**

yes

**Strengths And Weaknesses:**

Strengths
- A training-free intervention method to alleviate hallucinations.
- Evaluated on five benchmarks.
- Easy to follow.

Weaknesses
- Computational overhead, especially on larger models.
- The causal effects are estimated solely for individual heads, while during intervention the method applies top-k gating, which may introduce interaction effects among heads.
- Some recent training-free methods could be included for better comparison, such as [1].

[1] ONLY: One-Layer Intervention Sufficiently Mitigates Hallucinations in Large Vision-Language Models, ICCV 2025.

---

> ### Author Rebuttal · Authors · 2026-03-30
>
> ### W1. Inference overhead
>
> We thank the reviewers for raising the cost issue. We agree that CRG increases inference latency over standard decoding because it estimates route effects and performs conflict-aware intervention at inference time. However, this cost is already carefully quantified in the paper **(Table. 6 & 19)** and remains moderate among training-free baselines: under the same setup, CRG has 2.27× latency with essentially no extra peak GPU memory (14950 MB, 1.00×), versus 2.01× / 1.05× for VCD, 7.12× / 1.52× for OPERA, and 6.52× / 1.54× for HALC.
>
> **Appendix F** further shows that most overhead comes from one extra forward pass, not search or optimization. On LLaVA-1.5-7B, total latency is 139.2 ms/step, with the two standard forward passes taking 114.7 ms/step (82.4%) and the gradient plus CRG-specific computation only 12.7 ms/step (9.1%).
>
> For larger models, we agree that the overhead does not disappear, but the same low-memory design still applies because CRG backpropagates only through lightweight gate scalars while keeping model parameters frozen. **Appendix E.1** further shows that the method remains effective on stronger/larger LVLMs (e.g., Qwen2.5-VL-32B and LLaVA-NeXT-34B). We therefore view cost as a real but already quantified tradeoff, rather than an unaddressed weakness.
>
> ### W2. Cross-Gate Interactions
>
> We agree that jointly applying top-$k$ gates can introduce interaction effects among heads, especially across layers. Our current theory mainly justifies the local first-order signals used for ranking/selection, rather than fully characterizing all such higher-order interactions.
>
> In practice, however, CRG is designed to keep this risk limited: the intervention is sparse, text-only, and graded rather than uniformly hard, and the conflict-specific gate ranges further constrain its strength (mild for Conflict-A, stronger only for Conflict-B). These choices make large joint perturbations less likely. The current results also do not suggest that such interactions dominate in practice. Please also see our response to **Reviewer m64C (Q3)** for a more detailed discussion.
>
> ### W3 & Q3. More Comparison Results
>
> We thank the reviewer for pointing out ONLY (ICCV 2025). Using the publicly reported POPE results and averaging over the three standard splits, CRG is stronger on both model families: 87.71 / 86.61 vs. 85.03 / 85.49 on LLaVA-1.5, and 87.26 / 86.22 vs. 86.72 / 86.04 on Qwen (Acc./F1).
>
> | Method |   LLaVA-1.5   |    Qwen-VL    |
> | ------ | :-----------: | :-----------: |
> |        |   POPE Avg.   |   POPE Avg.   |
> | ONLY   | 85.03 / 85.49 | 86.72 / 86.04 |
> | CRG    | 87.71 / 86.61 | 87.26 / 86.22 |
>
> We also provide additional comparisons with other recent methods at the following **anonymous link**: [[link](https://anonymous.4open.science/r/crg-results/)].
>
> ### Q1. Hallucination Types
>
> We thank the reviewer for this important question. We agree that CRG targets a specific but practically important hallucination source, namely cases where textual priors override available image-grounded evidence under modality conflict. Thus, some other hallucination types do fall outside the primary target scope. At the same time, the scope is broader than object-only hallucination: our evaluation also includes MME, MMHal-Bench, and AMBER, which cover attributes/colors, relations, OCR-sensitive perception, and broader multimodal categories.
>
> Regarding imperfect visual perception, CRG is not intended to hallucinate missing visual evidence; when the visual representation is weak or incorrect, it tends to make the model more conservative by suppressing unsupported prior-driven guesses, rather than encouraging a confident but ungrounded answer. We therefore view this as a scope boundary rather than a contradiction to the method’s main claim. Please also see our response to **Reviewer cDXQ (Q1)** and **Reviewer m64C (Q1)** for a more detailed discussion of weak/ambiguous visual evidence and same-direction perception bias.
>
>
> ### Q2. Per-Token Intervention in Generation
>
> We thank the reviewer for this important question. For open-ended generation, CRG is not based on a single-token head identification that is then applied to the entire answer. Instead, it is recomputed at every decoding step: we define $\ell_t=\log p(y_t \mid x, y_{<t})$, estimate the route effects for the current token, apply the gate, and repeat at the next step. The final answer is thus shaped by a sequence of per-token interventions, not by one fixed attribution. For discriminative tasks such as POPE, MME, and binary AMBER, the final answer is simply the current Yes/No decision, so this issue does not arise.
>
> More broadly, CRG is a decision-time correction mechanism rather than a model of the full global reasoning process. It uses the current token as a practical local signal and should be viewed as a step-wise control method for autoregressive generation, not as a claim that one token-level estimate fully explains the final answer.

---

> > ### Author Rebuttal · Reviewer_XBUd · 2026-04-04
> >
> > Thanks for the rebuttal, which addresses most of my concerns.
> >
> > Q1: The rebuttal frames this issue as a scope boundary, which helps reduce my concern. I would encourage the paper to discuss this more in the final version.
> >
> > Q2: I should have made this point clearer. My concern was that a single current token may not reliably reflect the reasoning behind the final answer. For example, some intermediate tokens may serve as linguistic realization or fluency, so applying CRG to every single token could introduce bias unrelated to the final answer.
> >
> > Overall, I am inclined to raise my score to 5 after the rebuttal, depending on the further discussion.

---

> > > ### Author Response · Authors · 2026-04-04
> > >
> > > Thank you for the thoughtful follow-up and the encouraging feedback. We are glad that the rebuttal addressed most of your concerns, and we appreciate the opportunity to clarify the remaining points below.
> > >
> > > ### Q1. Clarifying the Scope Boundary of CRG
> > > Thank you for the helpful follow-up. We are glad this point is now clearer.
> > >
> > > We agree that CRG is designed for hallucination cases driven by conflict between visual evidence and language priors. We will make this boundary more explicit in the final version, and clarify that CRG is meant as a targeted and complementary intervention.
> > >
> > >
> > >
> > > ### Q2. Why Does CRG Check Every Decoding Step?
> > > Thank you for the helpful clarification. This is indeed a design issue we explicitly considered during method development, and we appreciate the opportunity to explain it more clearly.
> > >
> > > We agree that not every decoding token should be interpreted as equally reflecting the final grounded decision. For example, in a response such as “It is a lovely dog,” the visually grounded semantic decision is concentrated more on content-bearing tokens such as “dog,” whereas tokens like “It / is / a” are less directly tied to the grounded semantic content, even though they may still shape the subsequent decoding trajectory. This naturally raises the question of whether CRG should intervene only on a subset of semantically important tokens, rather than checking every decoding step.
> > >
> > > In fact, this concern was part of our design process. When developing CRG, we explicitly explored three practical intervention strategies on CHAIR using LLaVA-1.5-7B with max token length 128:
> > >
> > >
> > > (a) CHAIR-vocabulary-triggered CRG: intervene only when the current token matches the closed object vocabulary used by CHAIR, i.e., the MS COCO object categories and their synonym list. This is oracle-like because it uses benchmark-specific knowledge of which token types are relevant to CHAIR evaluation.
> > >
> > > (b) Step-wise CRG (ours): apply CRG at every decoding step, with the current-step route analysis determining which heads receive text-route gating and how strongly they are gated.
> > >
> > >
> > > (c) First-token-only CRG: intervene only at the first generated token. This setting is motivated by Zhao et al. (ECCV 2024) [1], who show that the first-token distribution can contain strong response-level signals, while such signals weaken in later tokens.
> > >
> > > | Setting | CHAIRs ↓ | CHAIRi ↓ | Recall ↑ | Efficiency  | Generalizable? |
> > > |---|---:|---:|---:|---|---|
> > > | (a) | 33.4 | 10.6 | 78.6 | High | No |
> > > | (b) | 34.2 | 11.2 | 77.8 | Moderate | Yes |
> > > | (c) | 45.3 | 14.2 | 77.1 | Very high | Yes |
> > >
> > >
> > > These results suggest three points.
> > >
> > > First, (a) is slightly better than (b). This is consistent with the intuition that if one had oracle knowledge of which decoding positions correspond to object tokens, more selective intervention could be marginally more precise.
> > >
> > > Second, the gap between (a) and (b) is small, which we believe is the key point here. This suggests that the most critical hallucination-relevant conflicts are concentrated at semantically content-bearing positions (e.g., object or attribute words), even though earlier or more fluency-oriented tokens may still influence the subsequent decoding trajectory. As a result, even though CRG is applied at every decoding step, its practical effect remains close to the oracle-style object-triggered variant.
> > >
> > >
> > > Third, (c) is still a meaningful and efficient baseline. Its reasonable recall suggests that the first token indeed carries useful coarse response-level information, consistent with Zhao et al. [1]. However, it still underperforms (a) and (b), indicating that first-token signals alone are insufficient, since hallucinated content often emerges at later decoding positions.
> > >
> > >
> > > Therefore, we agree that more selective triggering is an interesting future direction. However, the oracle-style variant (a) depends on benchmark-specific object vocabularies and is therefore not applicable to general open-ended generation or practical deployment. For this reason, our current design adopts a general step-wise decision-time mechanism: CRG is applied at every decoding step, while the current-step route analysis determines a selective, head-level, conflict-aware text-route intervention. We will clarify this design tradeoff more explicitly in the final version.
> > >
> > >
> > > [1] The First to Know: How Token Distributions Reveal Hidden Knowledge in Large Vision-Language Models? ECCV 2024.

---

### Official Review · Reviewer_m64C · 2026-03-21

**Soundness:** 3
**Presentation:** 4
**Significance:** 3
**Originality:** 3
**Overall Recommendation:** 5
**Confidence:** 3

**Summary:**

The paper introduces an inference-time method to reduce hallucination in VLMs by identifying the attention heads in the model where the textual and visual signals conflict, and gating the textual signal. The method introduces an intuitive metric, Visual Reliance Index (VRI), at each layer and head, that accounts for the do-effect of activating visual/textual gates on the final log-probs or yes/no margin. By estimating this effect using a single forward pass, the method efficiently bypasses the need for several (potentially thousands) of forward passes to identify the VRI on each head. Finally, the CRG method shows promising positive results in reducing hallucination across benchmarks like POPE, MME and MMHal-Bench.

**Compliance With Llm Reviewing Policy:**

Affirmed.

**Final Justification:**

The authors have addressed my weaknesses. Please refer to my rebuttal comment.

**Key Questions For Authors:**

1. The choice of only gating the textual effect is somewhat justified by the use-case of this work, but I think that there may be several situations where gating the visual effect may also be useful. For example, if a question is about an occluded object in the image, the textual pathway may do a good job of “guessing” that the occluded object is. If CRG then suppresses the pathway, it may yield worse outputs. Are there benchmarks or counterfactual cases, different from POPE, MME, etc. where applying CRG hurts performance?
2. In Sec. 4.1 - Conflict-specific gate ranges, how are gate ranges determined? Are they model-agnostic, and if so, why? Also, would it be possible for it to directly be a function of VRI rather than having author-determined ranges?
3. Table 5: It is interesting that conflict-A and conflict-B are complementary, and both methods yield nearly similar performance to the CRG (A+B) approach. If so, it might indicate that as these gates are applied, the impact of one gate may positively or negatively complement the impact of another gate in downstream layers. A negatively complementary selection of gates may unnecessarily hurt performance (e.g. one gate may reduce hallucination, but now the new latents computed may reverse how another gate downstream behaves). It could be interesting to have a discussion around this, or explain why this may not be the case.

**Limitations:**

yes

**Strengths And Weaknesses:**

- The justification for VRI in comparison to previously common Visual Attention Ratio for detecting the impact of each head on the model output is quite intuitive and clearly explained
- The main trade-off of this method is that do-effect over every head would require many forward passes, making it expensive, but the paper does a good job of justifying the single-forward pass estimate (with theoretical proof, as well as a small scale validation in the appendix) as a reliable alternative. With this, each inference requires 2 forward passes, but other hallucination mitigation approaches seem to require even more compute/memory (Table 6).
- Overall, the paper is well-written and easy to follow, with a strong motivation and good analysis.

---

> ### Author Rebuttal · Authors · 2026-03-30
>
> ### Q1. Weak/Ambiguous Visual Evidence
>
> Thank you for this important point. We agree that the reviewer’s scenario is valid: under weak visual evidence, such as occlusion or blur, the textual pathway can sometimes provide useful contextual guesses, and suppressing it may make CRG more conservative and occasionally hurt helpfulness or task performance.
>
> At present, we do not include a benchmark specifically designed to isolate such weak-evidence cases, so we cannot claim a systematic evaluation of when CRG may hurt. However, we do acknowledge this possibility qualitatively in the paper, and **Fig. 8(d)** provides an illustrative example: under degraded visual evidence, regular decoding gives an over-confident affirmative answer, whereas CRG responds more cautiously. Importantly, across the benchmarks we do evaluate, we do not observe a broad degradation trend; CHAIR recall is preserved or improved, and MME is broadly improved.
>
> Our decision to gate only the textual effect is therefore a task-driven design choice rather than a claim that textual priors are never useful. The goal of CRG is to mitigate unsupported hallucinations in grounded generation. Accordingly, when the image is unclear, we prefer the model to be conservative rather than make a prior-driven guess that is not visually supported. In this sense, CRG prioritizes reliability and grounded faithfulness over speculative helpfulness in ambiguous cases.
>
> ### Q2. Conflict-Specific Gate Ranges
>
> We thank the reviewer for this question. The current gate ranges are manually specified but model-agnostic: we use $(0.5,1.0)$ for Conflict-A and $(0,0.5)$ for Conflict-B. This is not because these exact numbers are uniquely optimal, but because they provide a simple, non-overlapping split of the normalized gate space into mild versus strong text suppression, matching the semantics of the two conflict types.
>
> They are shared across models because they encode conflict semantics rather than model-specific effect magnitudes. Most adaptation already comes from the selected layer range, head budget $k$, and the VRI-based rank schedule within each conflict type.
>
> A direct VRI-to-gate mapping is certainly possible and could be a useful refinement. Our current design is already partially VRI-dependent: we first select the smallest-VRI heads and then assign gates monotonically by rank within the chosen interval. We use the current range-plus-rank scheme as a simple and stable starting point, while a continuous or learned VRI-based mapping is an interesting direction for further improvement.
>
> ### Q3. Cross-Gate Interactions
>
> We thank the reviewer for this insightful point. We agree that the A/B interventions are jointly applied and can therefore introduce higher-order interactions, especially across layers. Within a layer, the effect is structurally closer to additive, because route outputs are first computed from the same baseline forward and only then selectively gated before linear aggregation. The main source of non-additivity therefore arises downstream, where an earlier intervention changes the hidden states seen by later gates.
>
> A useful local view is to write the decision score as a function of the selected gates $g$:
>
> $$
> \ell(g)-\ell(\mathbf{1}) \approx \sum_i \frac{\partial \ell}{\partial g_i} \bigg\rvert_{\mathbf{1}} (g_i-1)+ \frac{1}{2}\sum_{i\neq j}\frac{\partial^2 \ell}{\partial g_i \partial g_j}(\tilde g)\(g_i-1)(g_j-1)+\cdots
> $$
>
>
> Here, the first-order terms are exactly the local signals used by CRG for ranking/selection, while the reviewer’s concern corresponds to the cross terms, i.e., whether one gate changes the effect of another. We agree that our current theory mainly justifies the reliability of these local first-order signals, rather than fully characterizing all higher-order interactions.
>
> While our theory does not fully characterize these interactions, there are still several reasons to expect them not to dominate in practice. First, the intervention is sparse, which reduces the number of active gate pairs $(i,j)$. Second, the gates are text-only and graded rather than uniformly hard, which reduces the perturbation factors $(g_i-1)(g_j-1)$. Third, effective interventions concentrate in a contiguous mid-to-upper layer block (Fig. 6), which limits how far cross-layer perturbations can accumulate and makes large couplings less likely.
>
> Empirically, if strong negative complementarity were dominant, A+B would underperform a single-branch variant or show brittle sensitivity to $k$, $\gamma$, or the layer range. Instead, Table 5 shows that A+B is consistently best, while Figs. 6–7 show smooth sensitivity. We therefore view higher-order cross-gate interaction as a real but secondary limitation, with the closeness of A-only/B-only to A+B better explained by overlap and diminishing returns than by harmful interference.

---

> > ### Author Rebuttal · Reviewer_m64C · 2026-04-04
> >
> > Thank you for the rebuttal. All the points make sense, and I think that these are limitations that can be acknowledged in the paper and worked on in future works. Since my score is already an Accept, I'd like to keep it that way. Best of luck!

---

> > > ### Author Response · Authors · 2026-04-04
> > >
> > > Thank you very much for the thoughtful follow-up and for the positive assessment. We are glad that our rebuttal addressed your concerns. We will make sure the relevant limitations are clearly acknowledged in the final version. Thank you again for your helpful feedback and support.

---

### Decision · Program_Chairs · 2026-04-30

**Decision:**

Accept (spotlight)

**Comment:**

The paper introduces an inference-time approach for reducing hallucinations in VLMs. Proposed approach identifies attention heads in the model where the textual and visual information are in conflict and, in such cases, reduces the impact of textual (prior) information.

Paper received the following ratings: 3 x Accept, 1 x Weak Reject. Overall, reviewers agree that the approach is interesting, paper is well-written and easy to follow. Reviewer comments focused on (1) gating design and ablations [m64C], (2) ability of the model to mitigate only a certain type of hallucinations (where the perception is deemed more accurate) [XBUd, cDXQ], (3) lack of comparisons to recent methods (e.g., ONLY) [XBUd], (4) concerns regarding reliability of visual signals and preference of them over textual priors that can sometimes be helpful [cDXQ], (5) sensitivity to hyperparameter and head selection, and (6) applicability in the blackbox setting [cDXQ, zkFn].

Authors have provided a rebuttal that addressed many of these points. Generally reviewers felt that their concerns were well addressed. At the same time, [cDXQ] maintained that while some  concerns and questions were addressed, and the paper proposes a nice mechanistic way to resolve inconsistencies between visual and textual signals, the approach is still limited.

AC has carefully read the reviews, rebuttal and the discussion that followed. AC believes that the topic of the paper is important and timely, and the proposed  approach is interesting and meaningful, and agrees with majority of reviewers that the work would make a good contribution to ICML. For these reasons, the recommendation is to Accept the paper. Authors are encouraged to incorporate the rebuttal comments into the camera ready.